

# Referent seismic crustal model of the Dinarides

Katarina Zailac[1], Bojan Matoš[2], Igor Vlahović[2], Josip Stipčević[3]

[1] University of Zagreb University Computing Centre SRCE, Zagreb, 10000, Croatia
[2] Faculty of Mining, Geology and Petroleum Engineering, University of Zagreb, Zagreb, 10000, Croatia
[3] Department of Geophysics, Faculty of Science, University of Zagreb, Zagreb, 10000, Croatia

*Correspondence to*: Josip Stipčević (jstipcevic.geof@pmf.hr)

## Abstract

Continental collision zones are structurally one of the most heterogeneous areas intermixing various units within relatively small space. The good example of this are the Dinarides where thick carbonate complex cover overlain older crystalline basement units and remnants of subducted oceanic crust. This is further complicated by the highly variable crustal thickness ranging from 20 to almost 50 km. In terms of spatial extension, this area is relatively small, but covers tectonically very differentiated domains making the analysis complex, with significant challenges in areas with less data coverage.

Presently there is no complete 3-D crustal model of the Dinarides (and the surrounding areas). Using the compilations of previous studies, we have created vertically, and laterally varying crustal models defined on a regular grid for the wider area of the Dinarides, also covering parts of Adriatic Sea and SW part of the Pannonian Basin. In addition to the seismic velocities (P- and S-) and density, three interfaces in our model were defined – Neogene deposits bottom, Carbonate complex bottom and Moho discontinuity. Neogene deposits and the Paleozoic to Eocene Carbonate complex rocks are not present in all areas of the model whereas Moho discontinuity depth is defined for the entire model. The newly derived model has been compared with the simple 1D model used for routine earthquake location in Croatia, and it proved to be a significant improvement.

The model derived in this work represents the first step towards improving our knowledge of the crustal structure in the complex area of the Dinarides. We hope that the newly assembled model will be useful for all the forthcoming studies which require knowledge of the crustal structure.

## Introduction

First seismic investigation done in the wider Dinarides region can be traced to Mohorovičić's famous discovery of the boundary between Earth's crust and mantle (Mohorovičić, 1910). The earliest deep seismic sounding experiments investigating the crustal structure beneath the Dinarides were conducted in the 1960s (Dragašević and Andrić, 1975; Aljinović, 1983; Aljinović et al., 1987). The most important results from those early investigations were about thickness of the upper crust carbonate cover (Aljinović, 1983). In more recent times, there was another set of active seismic experiments, the ALP 2002 (Brückl et al., 2007) focused on the investigation of the Eastern Alps but also covered the northernmost part of the Dinarides and the Pannonian basin. As



part of the same international experiment, Šumanovac et al. (2009) modeled velocities in the crust and uppermost mantle from the measurements taken along the Alp07 profile located at the crossing between NW Dinarides and SW margin of the Pannonian basin. The results from the Alp07 profile were later extended by several studies including gravimetric modeling, P-wave receiver function analysis and local earthquake tomography (Šumanovac, 2010; Šumanovac et al., 2016; Orešković

et al., 2011; Kapuralić et al., 2019), covering only the NW part of the Dinarides and SW part of the Pannonian basin. These studies reported a two-layer crust under the Dinarides, and virtually one-layer crust in the Pannonian basin.

The first Moho map of the Dinarides was compiled by Skoko et al. (1987) utilizing gravimetric data from that area. Similarly, Šumanovac (2010) used results from active seismics (Alp07) to

calibrate gravimetric data in the Dinarides and get a more accurate map of Moho depths in the region. Stipčević et al. (2011) were first to use direct seismic measurements in the central and southern Dinarides extending the analysis of Moho depths to that region. For this they employed the P-wave receiver functions (PRF) method and by modeling it found an intra-crustal reflector in the Internal Dinarides. Stipčević et al. (2020) extended the PRF analysis by including significantly

more stations and created the map of Moho depths beneath the Dinarides. In that extended receiver function study, Stipčević et al. (2020) report significantly thicker crust in the southern Dinarides compared to the previous studies.

Inline with the crustal exploration there have also been some recent investigations of the uppermost mantle. Using the S-wave receiver functions (SRF) Belinić et al. (2018) mapped the lithosphere–

asthenosphere boundary (LAB) under the Dinarides. The most interesting feature of that LAB map is the lack of deep lithospheric root beneath the central Dinarides, which was interpreted as thinning of the lithosphere due to possible lithospheric mantle delamination. Two years later, there was a complementary study by Belinić et al. (2020), using the Rayleigh wave tomography to obtain the upper mantle S-wave velocity model for the greater Dinarides area. Similarly, to their first

work, authors report missing deep lithospheric root under the central Dinarides, with a deep high velocity anomaly visible in the south Dinarides.

From this short outline of the main geophysical investigations done in the wider Dinarides area it is obvious that the crustal structure of the region is fairly complex with quite a long history of geophysical exploration. Despite this, there has never been an attempt (as far as we know) to

combine these results to create an extensive regional 3-D crustal model for the Dinarides. The area was covered by the global and continental scale models, but the authors of these studies pointed out the lack of available data in the Dinarides area (e.g. Grad et al. 2009, Molinari and Morelli 2011, Artemieva et al., 2013). Furthermore, there have been recent seismic studies in the Dinarides, investigating both the crust and uppermost mantle, which shed light on previously

poorly studied parts of this region. In this study we will focus on the crust and include all the data regarding crustal structure available to us, into a 3-D model covering the area of the Dinarides. Besides the velocity as the main parameter describing the crustal structure, we have also included existing data on the Moho discontinuity depth, and the approximate Carbonate complex thickness





for the Dinarides. Even though our crustal model is focused on the Dinarides, part of it also covers
the SW margin of the Pannonian basin and Adriatic Sea.

**Tectonic and geological setting**

The Alpine–Carpathian–Dinaridic–Albanide–Hellenic orogenic system is a part of a Circum-Mediterranean orogenic system. Encircling the Neogene Pannonian Basin System (PBS), this orogenic system is constructed of tectonostratigraphic units derived from both Adriatic microplate
and European plate that were separated by Neotethyan ocean (Schmid et al., 2008, 2020, and references therein). Tectonic units were amalgamated by fold-and-thrust systems of different polarity facing either European foreland (e.g., Western and Eastern Alps, and Carpathians) or Adriatic foreland (e.g., Southern Alps, Dinarides, Albanides, Hellenides) because of change in subduction polarity between the European plate and Adriatic Microplate (Schmid et al., 2008,
2020; Ustaszewski et al., 2008).

The tectonic evolution of these large orogenic systems, i.e., Dinarides *sensu lato*, started with the Triassic (c. 220 Ma) opening of the Neotethys oceanic embayment between the African and Eurasian Plates. The Neotethys Ocean continued spreading during Late Triassic and Early to Mid-Jurassic, being interrupted only by intra-oceanic subduction of Western Vardar oceanic domain
and ophiolite obduction onto the eastern margin of the Adriatic Microplate (see Schmid et al., 2020 for details). According to Schmid et al. (2008) Neotethys Ocean, i.e. the Eastern Vardar oceanic domain remained open through the Late Jurassic–Cretaceous period (see Ustaszewski et al., 2009 and references therein).

The final closure of the Neotethys oceanic realm coincided with Adria Microplate and European
foreland collision during Late Cretaceous–Early Paleogene (Schmid et al., 2020 and references therein) that initiated formation of the Dinarides (Środoń et al., 2018). During the Middle Eocene–Oligocene the regional NE–SW oriented compression caused NE-directed continental subduction and formation of complex fold-and-thrust sheets in Dinaridic region due to SW-directed thrusting of Adria derived units, whereas in the internal Dinaridic domains, E–W oriented compression
caused formation of the W-directed thrusting of the Tisza Mega-Unit over the Internal parts of the Dinarides (e.g., Handy et al., 2019; Schmid et al., 2020; Balling et al., 2021 and references therein). Continuous convergence of the Adria indenter towards the European foreland in the Late Oligocene–Early Miocene further contributed to nappe stacking and thrusting in the Alps and Dinarides, but it also accommodated c. 400 km E-directed orogen-parallel lateral extrusion of the
ALCAPA block (including the Eastern Alps, West Carpathians and Transdanubian ranges north of the Lake Balaton) and active "back-arc-type" extension in the PBS (Royden and Horváth, 1988; Ratschbacher et al., 1991a, b; Frisch et al., 1998; Fodor et al., 1998; Tari et al., 1999; Csontos and Vörös, 2004; Horváth et al. 2006; Cloetingh et al., 2006; Schmid et al., 2008).

With an area of 80–200 km wide and nearly 700 km long, the Dinarides are a fold-and-thrust
orogenic belt constructed from thrust sheets divided into external and internal tectonic domains





(see **Fig. 1**). Both tectonic domains are comprised by lithological units formed on either proximal or distal portions of Adria Microplate that represents passive margin of the Adria Microplate or its Mesozoic carbonate platform (Adria Carbonate Platform, i.e., AdCP), respectively. As a result, the lithological succession in the Dinarides is characterized by Internal Dinaridic units that comprise 120 ophiolitic succession and pelagic derived sedimentary rocks, while External Dinaridic units are dominated by mainly Mesozoic shallow marine carbonate complex that in places reach c. 8000 m (Vlahović et al., 2005). Due to paleogeographic differences and tectonic complexity, Dinaridic lithological succession is spatially highly varied in its thickness and exposure (Vlahović et al., 2005, Schmid et al., 2020; Balling et al., 2021).

The oldest rocks cropping out in the External Dinarides are Carboniferous to Middle Triassic siliciclastic–carbonate deposits accumulated along the Gondwana passive continental margin, which due to regional extensional tectonics marked by Middle Triassic volcanism differentiated the area forming isolated shallow marine carbonate platforms (Vlahović et al., 2005). Through the Middle Triassic–Cretaceous timespan carbonate platforms in the region were surrounded by 130 deeper marine areas and characterized by mostly continuous shallow-marine carbonate deposition, though the last extensional phase in Toarcian resulting in disintegration in several smaller platforms, including Adriatic Carbonate Platform, Apenninic Carbonate Platform and Apulian Carbonate Platform (Vlahović et al., 2005 and references therein). During the 120 My of the AdCP's existence, i.e., from Early Jurassic to Late Cretaceous (locally even to early Paleocene – 135 Tešović et al., 2020), the thickness of deposited carbonate succession reached between 3500 and 5000 m (Vlahović et al., 2005 and references therein). The AdCP deposition ended due to regional emergence in the Late Cretaceous–Paleogene, which was linked and enhanced by tectonic deformations and continent collision of the Adria Microplate and European foreland yielding deposition of synorogenic flysch deposits in newly formed foreland (flysch) basins (Vlahović et 140 al., 2005; Schmid et al., 2008 and references therein). The final uplift of the Dinarides took place from the Middle Eocene to Oligocene (Dinaric phase; see Schmid et al., 2008; Balling et al., 2021 with references).

## Data

The main objective of this study is to create a referent seismic model of the crust beneath the 145 Dinarides. The area covered by this investigation is somewhat wider than the Dinarides, as it also covers the transition zones towards the Alps and Albanides, SW part of the Pannonian basin and parts of the Adriatic Sea (see **Fig. 1**). In terms of spatial extension, this area is relatively small, but covers tectonically very differentiated domains making the analysis complex, with significant challenges in areas with less data coverage.

Our basic approach was to create a one-layered crust with laterally and vertically variable parameters (seismic velocities and density). During data collection and integration within the new model we realized that the Neogene deposits in the Pannonian Basin make up a very thick and distinct layer that cannot be ignored, and just incorporating it into a one-layered model would be





an oversimplification. In the light of the Pannonian basin sedimentary complex being significantly
different from the rest of the crust we included a Neogene deposits layer on top of a laterally and
vertically varying crust. The same goes for the Mesozoic Carbonate complex in the Adriatic–
Dinarides region as there are numerous studies indicating distinct reflections from the bottom of
the Carbonate complex in the seismic reflection studies (Dragašević and Andrić, 1975; Aljinović,
1983; Aljinović et al., 1987). Given that we had no available data on any of the layer's parameters
(velocity or density) in the Carbonate complex, we only included the Carbonate layer depth in our
model. Here, velocity and density of the carbonate layer were obtained using the same velocity
(and density) data we had available for the rest of the crust. This choice seems reasonable, since
each of the profiles used crosses the part of the investigated area covered with carbonates, and
samples its features, even though none of the studies used explicitly interpreted carbonates as a
separate layer.

To compile the data for the new crustal model, we used all available and published results. Some
of them were not available in a digital form (mostly the studies published before 2010), and had
to be digitized manually. The datasets used are very diverse: including two 3-D crustal models
(one consisting of two-layered crust with interface depths while other had single-layered crust),
several 2D interface maps (Moho depths and Neogene deposits maps), three seismic
refraction/wide angle reflection profiles (which were the basic source for velocity data) and five
gravimetric profiles. Coverage of the datasets used to compile the 3-D crustal model of the
Dinarides is shown in **Fig. 1.** and details are listed in **Table 1**.

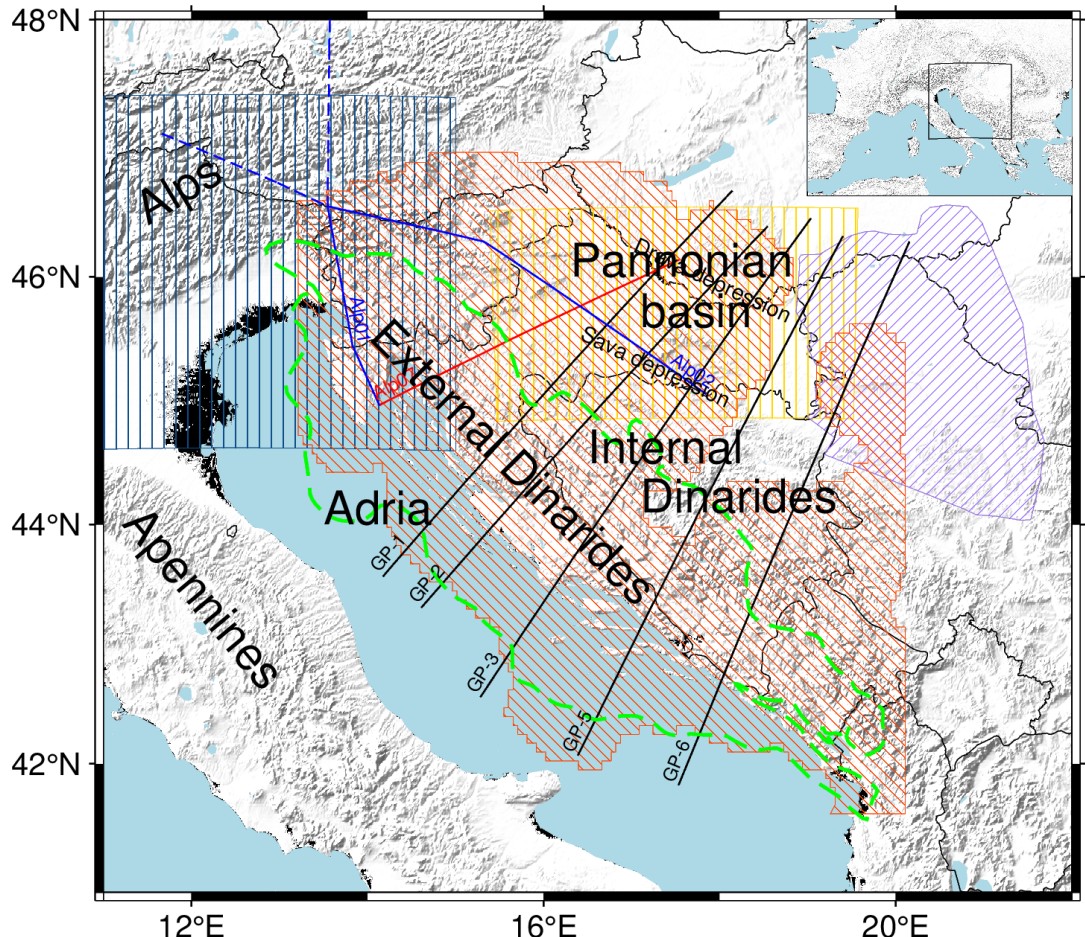

**Figure 1.** Compilation of data used in this study. Blue lines mark Alp01 and Alp02 profiles (Brückl et al., 2007) – only the full line parts are used in the study; the red line is the Alp07 profile (Šumanovac et al., 2009); black lines are gravimetric profiles GP-1 to GP-6 (Šumanovac, 2010). Orange shaded area shows Moho depth data coverage from the receiver function study of Stipčević et al. (2020), blue vertically shaded area is the NAC model (Magrin and Rossi, 2020). The yellow shaded area shows the extent of Saftić et al. (2003) data on the Pannonian basin Neogene deposits, and the purple shaded area shows the extent of data on the Pannonian basin Neogene deposits from Matenco and Radivojević (2012). Green dashed line marks the extension of the AdCP Carbonate complex from Tišljar et al. (2002). The sources of the data shown in this map are listed in **Table 1**.



**Table 1.** List of the data sources used for the Dinarides crustal model.

| Profile, model, or project name | Data type | Reference | How were data obtained | Parameter that data were used for |
|---|---|---|---|---|
| Alp01 and Alp02 | Seismic refraction and wide-angle reflection | Brückl et al. (2007) | profiles manually digitized | Vp, Moho |
| Alp07 | Seismic refraction and wide-angle reflection | Šumanovac et al. (2009) | profile manually digitized | Vp, Moho |
| GP-1, GP-2, GP-3, GP-5, GP-6 | 2D gravity modeling | Šumanovac (2010) | profiles manually digitized | Vp, density, Moho |
| | Receiver functions | Stipčević et al. (2020) | available in digital form | Moho |
| The Northern Adria Crust (NAC) model | Multiple data sets | Magrin and Rossi (2020) | available in digital form | Vp, density, Moho |
| | Isopach map | Saftić et al. (2003) | digitized manually | Neogene deposits bottom depth |
| | Multiple data sets | Matenco and Radivojević (2012) | digitized manually | Neogene deposits bottom depth |
| | Carbonate complex distribution | Tišljar et al. (2002) | digitized manually | carbonate bottom depth |
| EPcrust | | Molinari and Morelli (2011) | available in digital form | Vp, Moho |
| SRTM15+V2.0 | Global elevation grid | Tozer et al. (2019) | available in digital form | Topography |



There are three interfaces defined in the presented model – Carbonate complex bottom, Neogene deposits bottom, and Mohorovičić discontinuity (Moho). It should be noted that not all the interfaces are defined at all locations in the model (except for Moho). The parameters of the model (seismic velocities and density) are defined on a regular grid and were calculated differently for

the Neogene deposits layer and the rest of the crust. Since there was no velocity nor density information readily available for the Carbonate layer, its parameters were not assessed separately, as was the case for the Neogene deposits layer, but were calculated the same way as in the rest of the crust. The Carbonate bottom interface was kept in the model for the sake of completeness.

To define the interfaces, we needed the data on their depths. The information about the Moho was

compiled from many available studies. The main source of Moho data was the receiver function (RF) study of Stipčević et al. (2020). These data are digitally available as the crustal thickness map on a regular 8.3 km × 8.3 km grid (location shown in **Fig. 1** as an orange shaded area), and it includes error estimates on the same grid. We also included Moho data from seismic refraction and wide-angle reflection profiles (Brückl et al., 2007; Šumanovac et al., 2009; profiles' locations

shown in **Fig. 1** as blue and red lines, respectively), and from the gravimetric profiles (Šumanovac, 2010; black lines in **Fig. 1**). All profiles (both seismic and gravimetric) are available as figures in digital form but had to be digitized manually. Each profile was first georeferenced, and the interpreted Moho depths were digitized every 5 km along the profile's length. For the profile data there are no detailed error estimates but the authors report that the Moho interface was resolved to

at least ±2–3 km for refraction and wide-angle reflection profiles. There are no such estimates for the gravimetric profiles. Since the reported errors are only general, we decided to include the error estimates as described in Grad et al. (2009). The authors had a similar problem while building the Moho model but had much more input data to come up with reasonable error estimates. According to them, the error estimates for Moho obtained from the seismic profiles should be about 6% of

the Moho depth. That estimate nicely fits with the error estimates reported by Brückl et al. (2007) and Šumanovac et al. (2009) for refraction and wide-angle reflection profiles used in this study. As for error estimates in gravimetric profiles, Grad et al. (2009) reported somewhat higher errors, of about 20% of estimated depths. Since there was no information on errors for gravimetric profiles, we used estimates from Grad et al. (2009). For the NW part of our model, the Moho data

from the high-quality and digitally available Northern Adria Crust (NAC) model (Magrin and Rossi, 2020) was also included. NAC interfaces are defined on a regular ~ 5 km × 5 km grid with error estimations on the same grid (location shown in **Fig. 1** as blue shaded area).

The data for the Neogene deposits depth came from several geological studies. The most important region of the study area covered with loose Neogene deposits is the SW part of the Pannonian

Basin. As the basis for the definition of the Neogene sedimentary cover thickness in this region we used data from Saftić et al. (2003), which covers the southernmost part of the Pannonian Basin (yellow shaded area in **Fig. 1**). While preparing the data, we encountered the problem that missing deposits depth information for the eastern part of our study area is causing artifacts on the border



of our model after the interpolation. To mitigate this, we included data from the study by Matenco and Radivojević (2012) (purple shaded area in **Fig. 1**). Data from both studies were obtained from georeferenced isolines of the Neogene deposits depths – the isolines were digitized every 5 km, which gives the impression of random spatial sampling. We did not use interpolation in this step, in order not to introduce an additional interpolation error. For the error estimation we again turned to the study of Grad et al. (2009), where they estimated that the error for Moho should be about 15% for manually digitized maps. Since there was no error estimate beforehand, we decided to use the same percentage for Neogene deposits depth estimate. For the NW part of the study area, we included deposits bottom information from the NAC model. The area of the Dinarides does not contain a Neogene soft deposits cover, at least not of significant thickness, since the basement is overlain by a thick layer of Mesozoic Carbonate rock complex. Therefore, for the area of the Dinarides, we used the Carbonate complex distribution described in studies of Tišljar et al. (2002) and Vlahović et al. (2005) and marked it as a zero-Neogene deposits-thickness area, although there are locally some very restricted Neogene deposits formed within intramontane basins. Here, the carbonate distribution border was georeferenced and digitized manually (green dashed line in **Fig. 1**).

The interface with the least data at our disposal was the Carbonate complex bottom depth. The Carbonate complex bottom depth was estimated combining geological and structural data published in available Basic Geological Maps at the 1:100,000 scale with accompanying Explanatory Notes that cover entire Dinaridic area, as well as geological-structural data published in studies of Tišljar et al. (2002), Vlahović et al., (2005) and Balling et al., (2021). Based on the collected data, we determined the spatial extent of the Mesozoic–Paleogene Carbonate complex. Since the Carbonate complex represent a very distinctive layer in the Dinarides, we additionally estimated its thickness. Accordingly, assessment of Carbonate complex thickness was initially performed at the scale of each of more than 80 geological maps covering the study area, using thicknesses presented in geological columns on each map. Derived Carbonate rock complex thickness values were further analyzed and recalculated in respect to deformation styles and large-scale structural relations (described by Balling et al, 2021). This means that the several mapped carbonate nappe systems in the Dinarides that could reach total stacking thicknesses up to 12000 m, were in most cases not evenly spatially distributed, and often were either omitted or significantly reduced. Furthermore, this implies spatial complexity of Carbonate complex stacking thicknesses in Dinarides which were probably very unevenly enhanced from its initial thickness (c. 6000 m; see Vlahović et al., 2005 for details) due to inherited paleogeographic differences along the Adria Microplate passive margin, structural position of nappe systems in respect to active collision front, as well as variable strain rates and stress orientation during the Cretaceous–Paleogene Adria–Europe collision. At the same time, Dinaridic nappe stacking systems are well known in the central and southern part of the Dinarides where Carbonate rock succession is extremely thick (**Fig. 3c**), incorporating up to four smaller scale thrust sheets.



As for the physical characteristics, P-wave velocities were extracted from seismic refraction and wide-angle reflection studies (Brückl et. al, 2007; Šumanovac et al., 2009), densities from gravimetric profiles (Šumanovac, 2010), and P- and S-wave velocities and densities from the NAC model (Magrin and Rossi, 2020). We consider P-wave velocities as the best-defined layer property, since there is largest number of data sources for this parameter. As for the S-wave velocity, there was only data from the NAC model, which is just a border area of our model. Therefore, we did not interpolate S-wave velocity separately, but estimated it from the P-wave model.

The NAC model is defined on a regular grid and was easily included in our data set. The interpreted seismic reflection and refraction profiles (Brückl et al., 2007; Šumanovac et al., 2009) were digitized manually on a regular grid with the horizontal spacing of 5 km (along the length of each profile) and vertical spacing of 1 km. Since the gravimetric profiles are interpreted in terms of homogeneous layers (density in a layer does not change vertically nor horizontally), we applied a slightly different logic. We used 5 km spacing along the profile and instead of regularly digitizing the densities in depth, we only picked one point in each layer, in the middle of the layer, with assigned layer density. It seemed the most logical course of action, assigning the density value to the middle of the layer, to represent the entire layer. A simplified example of gravimetric profiles digitization is shown in **Fig. 2**.

We should also mention the error estimation regarding the velocity and density data. The NAC model (Magrin and Rossi, 2020) had reported parameter errors for each grid point and for interfaces, so these were simply included in our data set. Seismic refraction and wide-angle reflection profiles (Brückl et al, 2007; Šumanovac et al., 2009) had a general estimate of velocity error of ± 0.2 km/s and ± 0.1 km/s, respectively. In those cases, we simply assigned that globally estimated error to each digitized data point. In the case of the density data calculated from the gravimetric measurements there was no error estimate. Therefore, we had to use other results to help us quantify error for that dataset. Since the only other source of density data was the NAC model, and since we had no reason to believe that the gravimetric profiles we used (Šumanovac, 2010) are of a lower quality than those included in the NAC model, we assigned the maximum error estimate (to make a conservative estimate) from the NAC model to each of the points from the digitized gravimetric profiles.



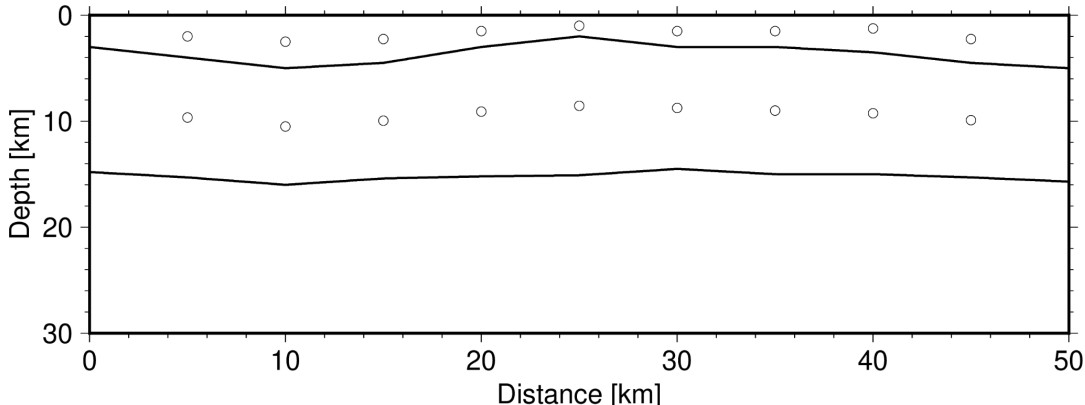

**Figure 2.** Simplified example of data sampling from one the gravimetric profiles with two interpreted layers. The dots represent digitized data points used to build our model. See text for details.

We used a regional EPcrust model (Molinari and Morelli, 2011) as the underlying model in order to fill the gaps in the data coverage. The EPcrust model is represented by three layers: sedimentary cover, upper crust, and lower crust, with a horizontal resolution of 0.5° × 0.5°. Each layer is characterized by laterally varying P- and S-wave velocity and density, and all the parameters are constant for each grid point. We include EPcrust Moho, Neogene deposits bottom depth and

velocity information only in parts with no other sources of data. This was done to remove interpolation artifacts in transition areas between the local data described before and the underlying EPcrust model. Therefore, we implemented a condition to include EPcrust data in our dataset: each grid point defined in the EPcrust model had to be distanced more than 100 km in each direction distant from the other data. That way, the data from the regional EPcrust model will not have too

much influence on more relevant, local data.

  For topography the SRTM15+V2.0 model (Tozer et al., 2019) was used. It is an updated global elevation grid at a spatial sampling of 15 arc seconds, and it also includes bathymetry. Since the grid of the SRTM15+V2.0 model is much more refined than the grid we used for the definition of our model (15 arc seconds in the topography model compared to about 3 arc minutes in our model),

we included it in our model by averaging all the values that fall inside our model cell.





**Model construction**

Ordinary kriging was used to interpolate all the interface data (Neogene deposits, Carbonate complex thicknesses and Moho discontinuity depths), and universal kriging to interpolate crust velocity and density data. Since kriging requires Cartesian coordinates, we transformed the data to ETRS89-extended/LAEA Europe[1] Cartesian coordinate system, which is defined on the entire investigated area. The transformations were done using the *pyproj* package (Snow et al. 2021). Kriging interpolation was done using the *gstat* package (Pebesma, 2004). Interface data were interpolated on a regular 5 km × 5 km grid, and the velocity and density were interpolated on a slightly more complicated grid: the horizontal grid is the same as for the interfaces (5 km × 5 km), but the vertical spacing changes depending on depth (in the first 10 km of depth, the spacing is 0.5 km; between depths of 10 km and 20 km, vertical spacing is 1 km, and at greater depths, vertical spacing is 2.5 km). This scheme was used to account for better sampling and more heterogeneous upper crust.

Initially, we specified a relatively large area between 10° and 20° east longitude and 39° and 48° north latitude as the starting region of investigation. We performed interpolation in the entire initial region for each interface separately. As mentioned in the previous section, most of the available data was related to the Moho discontinuity, and therefore the results of that interpolation are considered most accurate. Because of that, we used Moho discontinuity depth errors in combination with lower uncertainty areas of the Vp model as guidelines to mark edges of our model. The final model covers only the area between roughly 13° E and 20° E, and 42° N and 47° N.

Kriging does not allow multiple values at a single point in space (i.e., there is no overlapping). Therefore, we needed to handle the overlapping of data from various sources, before starting the interpolation. We tried to reduce subjectivity as much as possible, and therefore included known and estimated variances into the data processing. In case of the data overlapping, we calculated the value which were interpolated (depth for interfaces or velocity/density for layer parameters) at the given point as a weighted mean of multiple values from different sources, with inverse of variances used as weights:

$$\hat{d}_m = \frac{\sum_i \frac{1}{\sigma_i^2} d_{m,i}}{\sum_i \frac{1}{\sigma_i^2}},$$

where $d_{m,i}$ is the value at point $i$, and $\sigma_i^2$ is variance at point $i$.

We also included error estimates in the final model. To calculate the total error of the model, we followed the procedure applied by Magrin and Rossi (2020) for the derivation of the NAC model.

---

[1] https://epsg.io/3035



With the assumption of Gaussian distribution of errors, the total variance of the model is the sum
of two terms: the variance of the input data, and the variance from the interpolation itself. The
interpolation variance term was provided by the *gstat* package along with the interpolated data. To
calculate the variance at each grid point, we interpolated the input data variances on the same grid
as the data itself.

The right-hand side of **Fig. 3** (panels b, d and f) shows the standard deviations (the positive square
root of the variance) for Carbonate rock thickness, Neogene deposits thickness and Moho depth.
For the deposits bottom error estimation, we had the following models available: Tišljar et al.
(2002), Saftić et al. (2003), Matenco and Radivojević (2012), and Magrin and Rossi (2020) – the
rest of the area was filled with regional EPcrust data and is therefore of lower accuracy.

The most accurate source of velocity data was refraction and wide-angle seismic reflection profiles
(Brückl et al., 2007; Šumanovac et al., 2009), and the NAC model (Magrin and Rossi, 2020) but
as can be seen in **Fig. 1** these studies do not cover the central and southern area of the Dinarides.
Therefore, we also included velocities calculated using the values of density from gravimetric
profiles (Šumanovac, 2010). The differences in the digitization of the gravimetric profiles have
been described in the previous section. To calculate P-wave velocities from the available densities,
we used Brocher's (2005) empirical equation:

$$V_P = 39.128\rho - 63.064\rho^2 + 37.083\rho^3 - 9.1819\rho^4 + 0.8228\rho^5,$$

where $V_P$ is P-wave velocity in km/s, and $\rho$ is density in g/cm³. The equation is valid for densities
between 2 and 3.5 g/cm³ (with the correlation coefficient of ~0.999), the condition which was
satisfied for all the densities in gravimetric profiles used.

The velocity in the Neogene deposits is poorly known so it was estimated using Brocher (2008)
empirical relations. These relations account for increasing burden pressure, but not for variations
in other factors. Therefore, it is justified to use these relations as a first-order approximation,
because there are no similar relations readily available for the Pannonian Basin. We used Brocher's
Plio-Quaternary relations for the shallowest parts, and relations for Paleogene–Neogene
sedimentary rocks for the rest of the Neogene deposits.

**The main characteristics of the model**

After collecting and preparing the data as described in the previous sections, the next step was
interpolation. Ordinary kriging was used to interpolate model interfaces and universal kriging
when interpolating the layer properties (P-velocity, density) as the layer properties are distinctly
linearly dependent on depth. After interpolation, we smoothed Moho discontinuity and layer
properties but skipped the smoothing in case of the Neogene deposits and Carbonate complex layer
interfaces. The smoothing in those cases was omitted because the data used in derivation of those
interfaces came from similar sources, and smoothing would conceal known structures due to their



relatively small spatial dimensions (e.g. Sava and Drava depressions would have been concealed
        with smoothing). Model uncertainty was estimated as the sum of two factors: uncertainty in the
        input data and uncertainty from the interpolation. The estimation of the former is described in
        detail in previous sections, and the latter is available from the output of kriging.

        Interfaces embedded within the model are shown in the left-hand side of **Fig. 3** (panels a, c and e).
The shaded areas mark the region where the model is not well defined. Neogene deposits and
        Carbonate bottom depths have not been smoothed, since they have been assembled from a small
        number of equivalent sources. Moho discontinuity, on the other hand, is assembled from a variety
        of different sources, and therefore was smoothed using a Gaussian filter with a 100 km width.

        Since the model is mostly concentrated on the Dinarides where the topmost cover is predominantly
made of Mesozoic–Eocene carbonate rock complex, Neogene sedimentary cover is negligible for
        the large part of the model, but areas to the east and northeast of the Dinarides (i.e., SW Pannonian
        Basin) have thick Neogene deposits reaching thicknesses of more than 4 km. Most prominent
        features, clearly seen as two red bands in **Fig. 3a,** are the Sava and Drava depressions, with Drava
        depression being slightly deeper.

Carbonates bottom depth model (**Fig. 3c**) is almost a mirror image of the Neogene deposits bottom
        model. That was expected, since in the SW Pannonian Basin, where Neogene deposits are the
        thickest, underlying carbonate layer is thin, and vice versa – in the Dinarides, where the carbonate
        layer is the thickest, there are no prominent Neogene deposits. Carbonate thicknesses are well over
        5 km in the northern part of the Dinarides and they are getting even thicker going southwards along
the Dinarides chain strike (reaching cumulative thicknesses of almost 15 km in the southern part
        of the mountain chain). In the Adriatic Sea area, carbonates are thinning out going southwestwards,
        but that may also be partly caused by the relative lack of available data in that part of the model.

        As corroborated before, Moho discontinuity depth is the best constrained feature of the model.
        Greatest Moho depths in the investigated region are found in the SW part of the Dinarides
mountain chain, where it reaches depths of over 45 km (see **Fig. 3e**). To the NE, along the External
        Dinarides mountain chain strike Moho becomes shallower, reaching depths of around 40 km. In
        the SW part of the Pannonian basin crustal thickness is between 20 and 30 km, becoming even
        shallower going further east. In the Adriatic Sea (within the part covered with our model), Moho
        is shallower than in the Dinarides, but deeper than in the SW Pannonian Basin, with crustal
thicknesses between 30 and 35 km. At the transition from Adriatic Sea to the Dinarides mountain
        chain, the Moho depth change is gradual, whereas going towards the SW Pannonian basin from
        the Dinarides, the change is rather abrupt. This can be better seen in **Fig. 5**, which shows three
        profiles laid almost perpendicularly to the strike of the Dinarides.

        The right-hand side of **Fig. 3** (panels b, d and f) shows interface uncertainties (Neogene deposits,
Carbonate bottom and Moho depth, respectively). Given that a significant contribution to the
        uncertainty value is the uncertainty from the interpolation itself (which is of greater value at grid





points further away from the input data points), one can distinctly see the areas with less data coverage as areas with higher uncertainty. Moho depth uncertainty (**Fig. 3f**) is low in the entire area of interest, i.e., the wider Dinarides region. For Neogene deposits bottom the area of lower

data coverage is in the eastern part of the Internal Dinarides where there is less information available on sedimentary thickness (see also **Fig. 1**). On the other hand, that part of the Internal Dinarides is mostly covered by the exposed bedrock largely composed of low-grade metamorphic Paleozoic–Mesozoic formations with thin cover of Mesozoic carbonate rock complex (e.g., Schmid et al., 2008), so Neogene deposits thickness values here are mostly negligible. For

Carbonate rock layer thickness, the area of least accuracy is in the NE part of the investigated area (junction zone between Dinarides-Pannonian Basin - Southern Alps). Like the previous case, here the low accuracy is due to the lack of measurements on carbonate thickness as the region is covered by a thick layer of Neogene deposits.





**Figure 3.** Model interface depths and corresponding uncertainties: (a) Neogene deposits bottom depth, (b) Neogene deposits bottom uncertainty, (c) Carbonate bottom depth, (d) Carbonate bottom uncertainty, (e) Moho discontinuity depth, and (f) Moho depth uncertainty.



The seismic velocity distribution within the model is depicted in the left-hand side of **Fig. 4** at four depths (5, 10, 20 and 30 km) showing most prominent features of the model crustal structure. At a depth of 5 km (**Fig. 4a**), the P-wave velocity in a large part of the External Dinarides is about 6 km/s. Given that we had no separate estimate for the velocity for the carbonate layer, we cannot discern it as a separate layer just looking at velocity values. In the rest of the model, one can see

that the velocity in the SW Pannonian Basin at depth of 5 km is slightly lower (just below 6 km/s) than in the rest of the investigated area. At a depth of 10 km (**Fig. 4c**) the velocity values in the SW Pannonian Basin are considerably lower than in the rest of the model, with values around 6 km/s. In the area of the Internal Dinarides, velocity at 10 km depth is slightly higher than in the External Dinarides and beneath the Adriatic Sea. It is hard to discern if this reflects the actual

structure, or if it is the consequence of the higher uncertainty in that part of the model (the velocity here was estimated from the density values from the gravimetric profiles, given that there were no other data sources available). Similar situation can be seen in **Fig. 4e** for a depth of 20 km. For this depth the velocities in the SW Pannonian Basin are reaching values above 6 km/s whereas values for the Dinarides are again higher (especially in the internal part) than in the rest of the investigated

area, with values above 6.5 km/s. In the lower part of the crust (**Fig. 4g**) at a depth of 30 km in the central part of the Dinarides the P-velocity values are reaching 7 km/s. In the same image the mantle velocity values are shown in grayscale due to the considerable difference between crust and mantle values and the fact that the crustal thickness in SW Pannonian basin is mostly less than 25 km. The mantle velocity variations are better seen in profile sections (e.g., see **Fig. 5f**). At the

30 km depth, the velocity is much higher in the south External Dinarides and below the Adriatic Sea (at least the part covered with our model) than in the northern External Dinarides. The mantle velocity shown here is not estimated as part of this model, but was taken from Belinić et al. (2020), by estimating it from the Vs model reported there using the standard P over S-velocity ratio for the upper mantle.

**Fig. 4** also shows error estimates for P-wave velocity at four depths: 5 km, 10 km, 20 km and 30 km (panels b, d, f and h, respectively). The lowest estimates are in the area where we used the NAC model input data (Magrin and Rossi, 2020), and in areas where data from active seismic profiles were available (Brückl et al., 2007; Šumanovac et al., 2009). The disposition of the errors shown in Fig. **4** was expected given the fact that the digital NAC model and the active seismic

profiles are of highest quality and that gravimetric data has higher uncertainty. Estimated uncertainty is highest in the area where gravimetric profiles (Šumanovac, 2010) are the main source of the data used to estimate P-velocity.







**Figure 4.** Velocity model depth slices and corresponding uncertainties: (a) model at a depth of 5 km, (b) uncertainty at a depth of 5 km, (c) model at a depth of 10 km, (d) uncertainty at a depth of 10 km, (e) model at a depth of 20 km, (f) uncertainty at a depth of 20 km, (g) model at a depth of 30 km, and (h) uncertainty at a depth of 30 km. In the panel (c) the positions of the profiles shown in **Fig. 5** are marked**.** The areas of lower resolution are shaded, and the gray color scale corresponds to the mantle velocity (see text for details).

**Fig. 5** shows depth variations of velocity along the profiles (locations marked in **Fig. 4b**) crossing the Dinarides perpendicularly with one profile (FF)' running parallel to the main strike of the mountain chain. The profile AA' (**Fig. 5a**) crosses the Dinarides in their northern part. The maximum Moho depth below the Dinarides on this profile (around 40 km) is the lowest compared to the other profiles. At a distance of ~270 km, the profile reaches the SW Pannonian Basin, which can be clearly seen as the thinning of the crust (between 20 and 30 km) and by the topmost layer of Neogene deposits of lower P-velocity. Although the Carbonate complex layer thickness is indicated with the dashed line, one cannot recognize it by looking just at the velocity values. Generally, the velocities in the part of the profile crossing the Dinarides are larger than in the part of the profile crossing the SW Pannonian Basin. The velocity gradually increases with depth, reaching values of about 6.7–7.0 km/s in the deepest part of the crust, apart from the SW Pannonian Basin, where the velocity just above Moho is lower, about 6.5 km/s.

The maximum Moho depth seen on the profile BB' (**Fig. 5b**) is somewhat greater than in the profile AA', a little over 40 km in the part of the External Dinarides. It is shallower in the Adriatic area (in the first 50 km of the profile) and in the Internal Dinarides (after about 220 km). At the very end of the profile, where it reaches the SW Pannonian Basin, one can see the same feature seen in the profile AA': thinner crust and slightly lower velocity just above Moho than in the rest of the profile. The Carbonate layer thickness, indicated by the dashed line, in the part of the profile crossing the part where Moho is the deepest, almost perfectly coincides with the velocity value of about 6.3 km/s. That feature can also be observed on other profiles (CC' and DD') at similar locations.

The profile CC' (**Fig. 5c**) crosses the Dinarides in their central part. Here the maximum Moho depth is almost 50 km. The profile reaches the SW Pannonian Basin only in the last 100 km, but it covers much of the Internal Dinarides. The Carbonate layer is of uniform thickness along the part of the profile covering the Dinarides (after the first 100 km, which cover the Adriatic area). The crust is thickest beneath the External Dinarides and is becoming thinner going both towards the Adriatic Sea and the Internal Dinarides. In this central part of the External Dinarides, there is also somewhat higher velocity recorded deeper in the crust, 7.0–7.2 km/s, just above the Moho. In the Internal Dinarides (between 250 to 300 km from the start of the profile), the velocity just above the Moho is a little lower than in the external part, around 6.7–7.0 km/s. As noticed in the previous two profiles, in the SW Pannonian Basin, the velocity just above Moho is even lower than in the Internal Dinarides. Similarly, as for the profile BB', the bottom of the Carbonate layer coincides



with the velocity values of about 6.2–6.3 km/s, except in the very beginning of the profile (first ~50 km). In this profile (but see also profiles DD' and EE') it is worth mentioning that the higher velocity body mapped in the upper mantle (Belinić et al. 2020) coincides well with the thickest

crustal section. This can be linked with the remnants of the subducted lithosphere and the ongoing underthrusting or lithospheric delamination (see e.g., discussion in Stipčević et al. 2020)

Profiles DD' and EE' (**Fig. 5d and 5e**) cross the southern part of the Dinarides. Here, the Moho reaches depths of over 50 km. Also, the crustal velocity at those depths is largest of all the profiles, reaching almost 7.5 km/s. Greater Moho depths and crustal velocity change can be best seen in the

profile FF' (**Fig. 5f**) running parallel to the Dinarides from northwest to southeast. In this profile Moho depth increases from around 40 km in the northern part to over 50 km in the southern part. Also, the crustal velocity just above Moho changes from just below 7.0 km/s in the north to almost 7.5 km/s in the south. Similarly, velocity in the mid-crustal zone (~20–25 km depth) is somewhat lower in the northern part of the profile, a little over 6.5 km/s, but becomes higher in the SE part

reaching values of about 7 km/s.









**Figure 5.** Profiles shown in **Fig. 4b**: (a) AA'; (b) BB'; (c) CC'; (d) DD'; (e) EE', and (f) FF'. Parallel full lines running approximately along the profile are the velocity gradient lines. The depth of the Carbonate layer as derived from known geological data is indicated as the dashed line close to the surface.

As mentioned, before it is interesting to note that the trend of velocity in the lower part of the crust corresponds to the velocity trends in the uppermost mantle. There is a positive velocity anomaly in the part of the uppermost mantle right below (or slightly offset from) the part of the crust with the deepest Moho. These positive anomalies in the work of Belinić et al. (2020) have been interpreted as a signal of the subducting Adria Microplate. Our model is mere interpolation of what was already known, but perhaps what we see here is part of the Adria crust being dragged along the uppermost part of the mantle being subducted below the Dinarides. As can be seen in the **Fig. 5a**, which is crossing into the SW Pannonian basin, the crustal velocity in that part is much lower than in the Dinarides, a feature also observed by Šumanovac et al. (2009). Perhaps the lower velocity is a feature of the Pannonian crust, whereas the relatively higher crustal velocity is a feature of the Adria crust. To make a definitive conclusion, more investigation should be performed.

From the smaller data set which includes the NAC model (Magrin and Rossi, 2020) and the gravimetric profiles (Šumanovac, 2010), we have interpolated the density values for the entire crust. The result is shown in **Fig. 6**. Keep in mind that for this interpolation there were only two sources of data, one of which had densities defined as isotropic layers (Šumanovac, 2010). Therefore, this parameter is much less accurate than P-wave velocity. As expected, this parameter reflects the results of Šumanovac (2010), since that was the main source of data. The density is slightly higher in the Internal Dinarides than in the External Dinarides for all the depths considered here, possibly coinciding with higher density crystalline crust. In the SW Pannonian Basin, the density has much lower values.



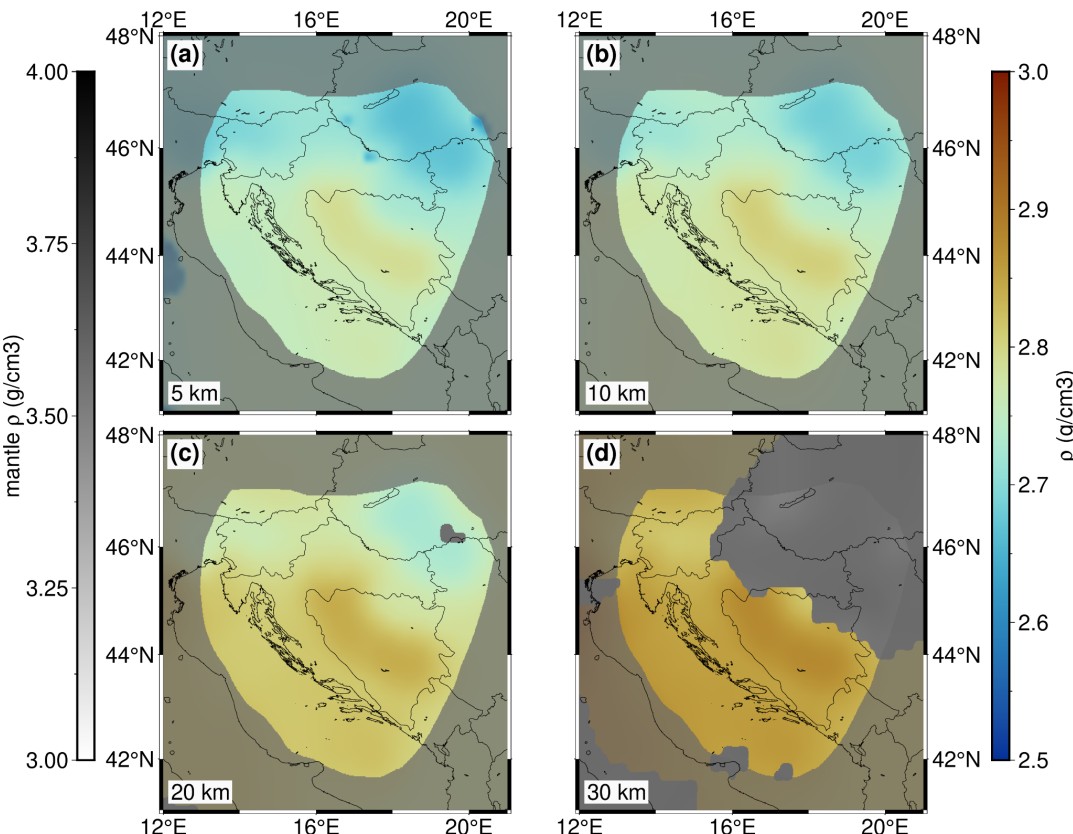

**Figure 6.** Density model depth slices for (a) 5 km, (b) 10 km, (c) 20 km, and (d) 30 km depth. The areas of lower resolution are shaded, and the gray color scale corresponds to the mantle density (see text for details)







**Fig. 7** shows the S-wave velocity at four depths. In this case, there was no measured S-wave
velocity data (the only available Vs results were from the NAC model which covers the western
corner of our study area) so these values were estimated using the P-wave velocity and Brocher
(2005) empirical relation.

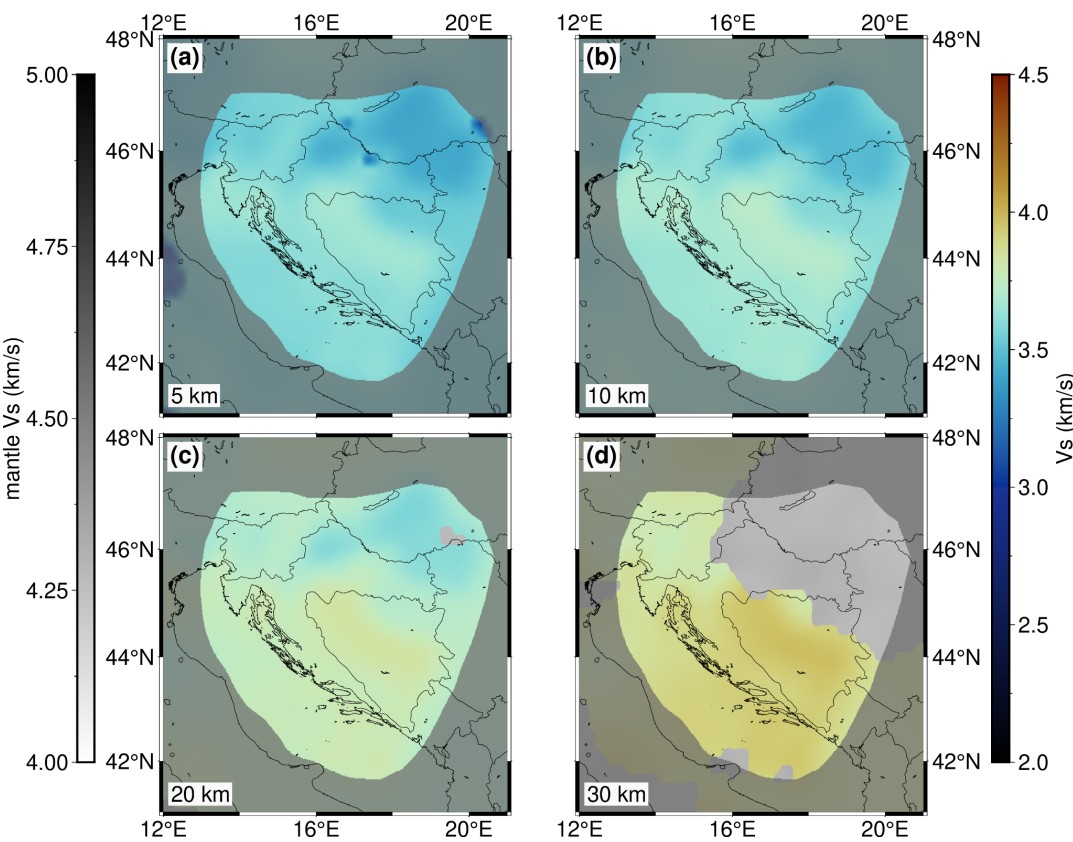


**Figure 7.** S-wave velocity depth slices for (a) 5 km, (b) 10 km, (c) 20 km, and (d) 30 km depth. The
areas of lower resolution and the gray color scale corresponds to the mantle velocity (see text for details).



**Discussion**

The creation of the presented 3-D model was inspired by the need for the more complete seismic model of the Dinarides. Although there have been previous studies estimating various properties of the crust in the region, the complete seismic model of the Dinarides crust and upper mantle still does not exist. In this study we assembled data from previous studies to create a first comprehensive model of the crust for the wider Dinarides area. Moho depth is the best constrained

parameter of our model, since there were enough high-quality data regarding this parameter. It is mainly based on Stipčević et al. (2020) study and our model reflect all the features noted therein. As shown in **Fig. 5f** the Moho deepens going from the NW to the SE along the main axis of the Dinarides. Also, from the profiles in **Figs. 5a–e** the change in Moho depth going from the Adriatic Sea towards the Dinarides mountain chain is much more gradual than on the other side, going from

the Dinarides towards the SW Pannonian basin, where this change is much more abrupt, almost step-like. The same feature has been observed by Šumanovac et al. (2009) and Šumanovac (2010).

In the case of Neogene deposits thickness the data used came from two sources (Saftić et al., 2003 and Matenco and Radivojević, 2012), and they provided adequate coverage in the SW Pannonian Basin where the thick Neogene deposits are located. Even though we used manually digitized

maps, therefore having less precise data, in the end, this parameter was adequately presented in our model. The thickest Neogene sedimentary cover can be found in the Sava and Drava depressions with thinner cover in the rest of the SW Pannonian basin and almost non-existent in other regions, most notably in the Dinarides, as well as in some hilly areas of the SW Pannonian Basin.

For P-wave velocity, the most valuable data available were the seismic refraction/reflection profiles (Brückl et al., 2007; Šumanovac et al., 2009) and the high-quality NAC model (Magrin and Rossi, 2020). As can be seen in **Fig. 1**, all the high-quality data are concentrated in the NNW part of the study area. In the southern part, we relied on the inverted gravimetric profile data (Šumanovac, 2010), which is not the ideal data source due to the high uncertainties and lower

resolution. Nevertheless, given the lack of other data sources for South Dinarides, even the data from the gravimetric profiles proved to be of high value. In the **Fig. 5a**, the northernmost profile shows the similar features as the profile interpreted by Šumanovac et al. (2009) – the crustal velocity in the SW Pannonian basin is much lower than in the Dinarides. The other profiles, located further south, also cutting across the Internal Dinarides, show that the crustal velocity in the

Internal Dinarides, is generally a bit higher than in the External Dinarides with relatively quick transition to the lower seismic velocity values in the Pannonian basin (see profiles CC', DD' and EE').

The velocity estimation for the Neogene deposits and the Mesozoic carbonates proved particularly challenging since there is little available data about this parameter. In the case of Neogene deposits,

we used Brocher (2008) relation for the deposits of similar age. For the carbonate layer, we could not derive any velocity–depth relation due to the lack of available data, so in this case, we simply





used the same velocity interpolation as for the rest of the crust. It seems, though, that at least in some parts of our new model, the Carbonate bottom depth coincides with the velocity of around 6.5 km/s.

To test how well the newly derived 3-D model represents the true structure, we calculated the travel times for a regional earthquake recorded on representative seismic stations in the wider Dinarides area (**Fig. 8**). We also calculated travel times using the simple 1D model with two isotropic crust layers currently employed for routine earthquake locating in Croatia. The 1D model's topmost layer is characterized by P-velocity of 5.8 km/s, and the deeper crustal layer has

the P-wave velocity of 6.65 km/s. For the same model the uppermost mantle velocity is 8.0 km/s. We then compared the travel times from both models with the true measured travel times. We used the Pn and Pg phases of the 2020 Petrinja Mw6.4 earthquake. The location of the earthquake and the stations that recorded the wave onsets are shown in **Fig. 8**. For the same stations we calculated the travel times using the 1D and the new 3-D model. For travel time calculation we used the Fast

Marching Method (de Kool et al., 2006) as implemented within the FMTOMO package (Rawlinson and Urvoy, 2006).



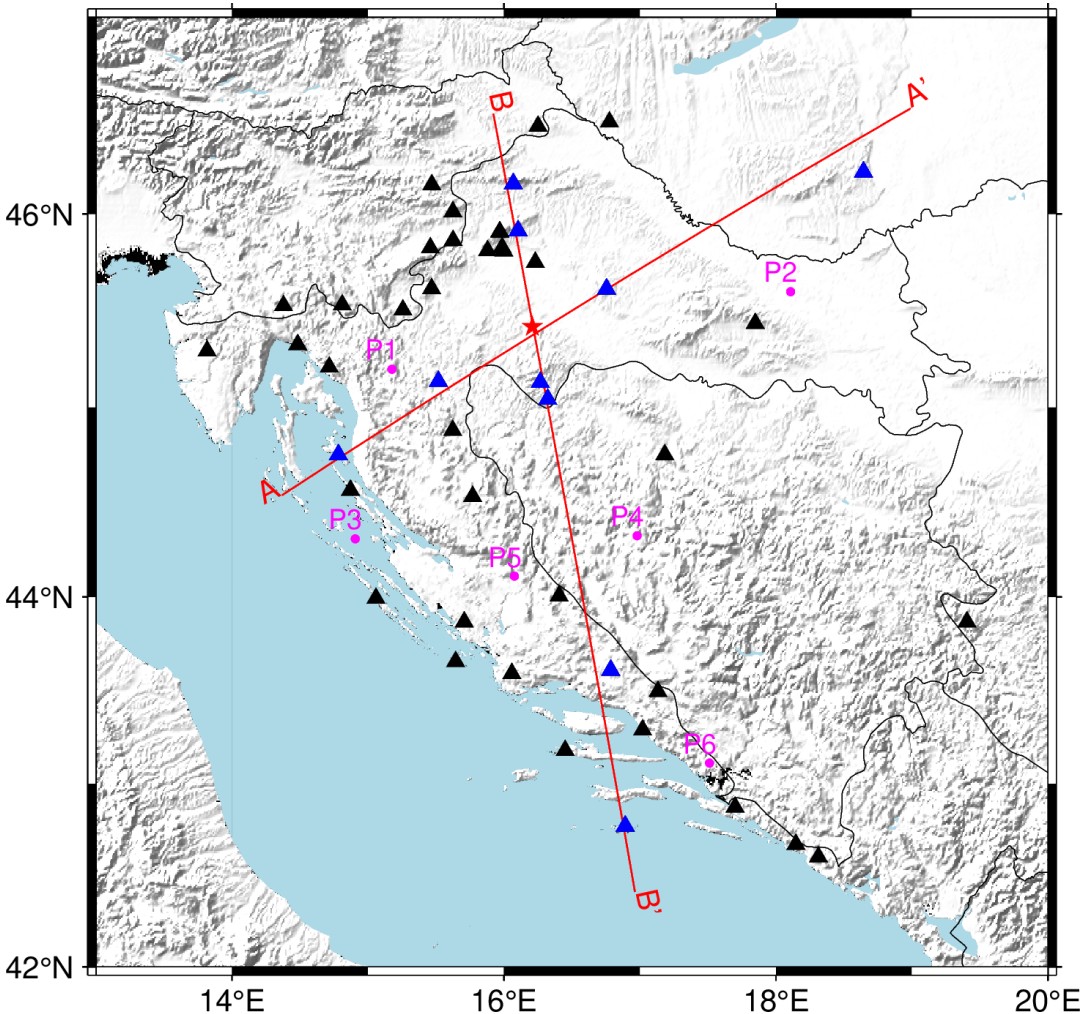

**Figure 8.** A map showing the epicenter of the Petrinja 2020 earthquake sequence mainshock (red star) and stations we used for calculation of traveltimes (black and blue triangles). Travel times for all the stations are shown in **Figs. 9** and **10**. The red lines mark the positions of the cross sections shown in **Fig. 11** (section AA') and **Fig. 12** (section BB'). The stations colored blue are the ones shown in **Fig. 11** and **Fig. 12**. The points colored in magenta mark the position of 1D models shown in **Fig. 13**.

Calculated travel times are shown in **Figs. 9** and **10**, for Pg and Pn phase, respectively. Figures show the differences in travel times calculated by the models (both 1D and 3-D) and observed travel times. When looking at Pg phases (**Fig. 9**), we can see improvement in calculated travel time accuracy when using the 3-D model for epicentral distances smaller than 50 km and over 100 km. For smaller epicentral distances, the more accurate travel times in the 3-D model relate to better specification of Neogene sedimentary cover with low P-wave velocity. On the other hand, for epicentral distances between 50 and 80 km 1D and 3-D models travel times are similarly offset





compared to the observed travel times, with times calculated using the 1D model being slightly more accurate. We believe this is due to the less accurate velocity sampling in the upper crust in the transitional zone between Internal Dinarides and Pannonian basin and lack of knowledge about spatial coverage of the Carbonate complex layer in this area. For greater epicentral distances we can see that travel times calculated using the 3-D model are much more accurate compared to those
calculated using the 1D model. That means that the crustal velocity derived in our 3-D model is a considerable improvement of the simple 1D model.

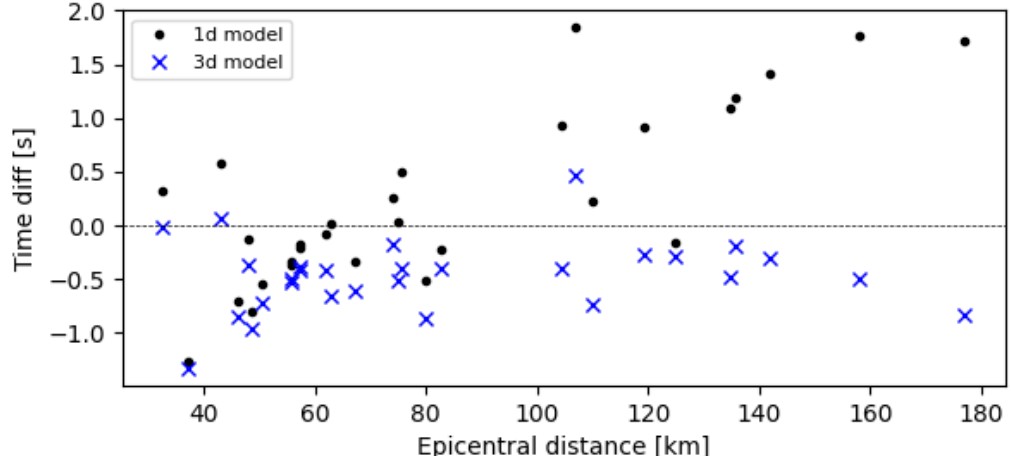

**Figure 9.** Pg phase travel times for the 2020 Petrinja earthquake: time difference between 1D model and observed travel times (black dots) and between 3-D model and observed travel times (blue crosses).

Concerning the Pn phases (**Fig. 10**), we can see that the travel times calculated using the 3-D model are generally closer to the actual observed travel times for all the epicentral distances shown than those calculated using the 1D model. In case of Pn phases, the uppermost mantle velocity plays a great part in the total travel times, so both the crustal model we derived here and the mantle model from Belinić et al. (2020) show improvement compared to the 1D model. There is still room for
improvement in the uppermost mantle velocity, since the model of Belinić et al. we used here is most accurate for greater depths (80–100 km).




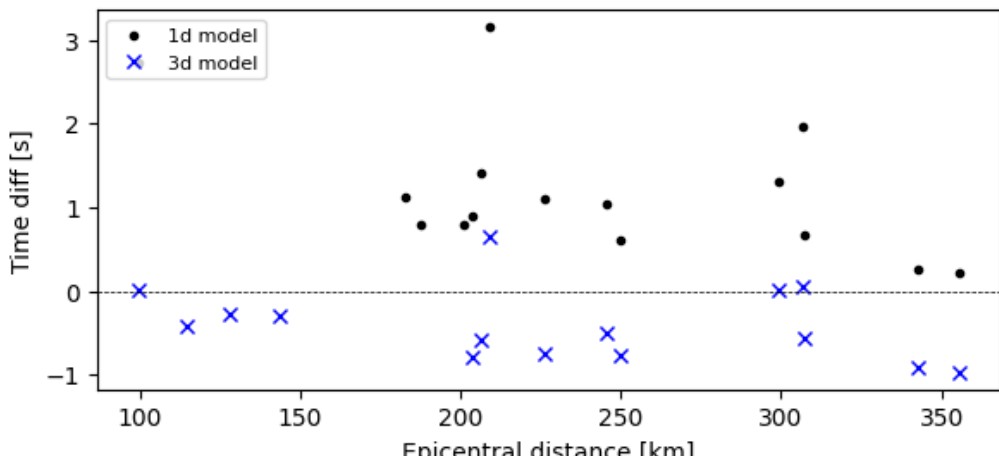

**Figure 10.** Pn phase travel times for the 2020 Petrinja earthquake: time difference between 1D model and observed travel times (black dots) and between 3-D model and observed travel times (blue crosses).

**Figure 11** shows a cross section AA' (from **Fig. 8**) of the newly derived 3-D model, with the calculated ray paths using the simple 1D and 3-D models. The section was chosen in a way to show travel times for both Pg and Pn phases, and we also tried to position it in such a way that it crosses almost perpendicularly the main strike of the Dinarides. There is also another cross section shown in **Fig. 12** (BB'), oriented approximately north to south. Position of the stations shown in

cross section BB' is also marked in **Fig. 8**. From both profiles the seismic rays cover completely different paths depending on whether they were calculated using the 1D or the 3-D model. For example, the Pg phases, when calculated using the 3-D model, travel much deeper than their 1D counterparts. Also, since the Moho in the Pannonian Basin of our 3-D model is much shallower than the Moho in the simple 1D model, the calculated rays using either 1D or 3-D model travels

on very different paths in the uppermost mantle. That is particularly visible in **Fig. 12**, in case of the ray path between the source and the most distant station shown. The ray path calculated for the 3-D model reaches depths of almost 60 km, while the same ray path calculated for the 1D model reaches depths of only 40 km, which is a huge difference. Given all that, it can be concluded that precise knowledge about the crustal model is very important for all seismic applications.



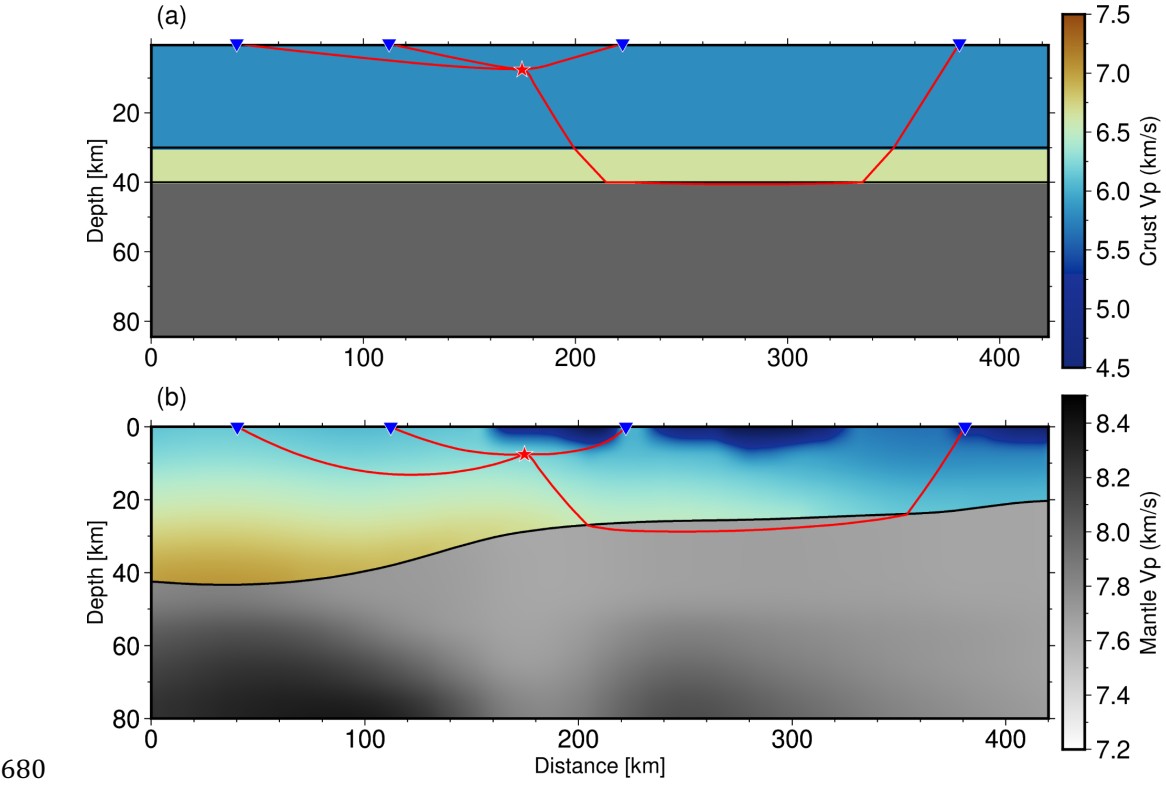


**Figure 11.** Earthquake ray paths in cross section AA' (for location see **Fig. 8**) for two models: (a) a simple 1D model with two isotropic crustal layers, and (b) our 3-D model with one anisotropic crustal layer. Colorbars are the same for both panels.



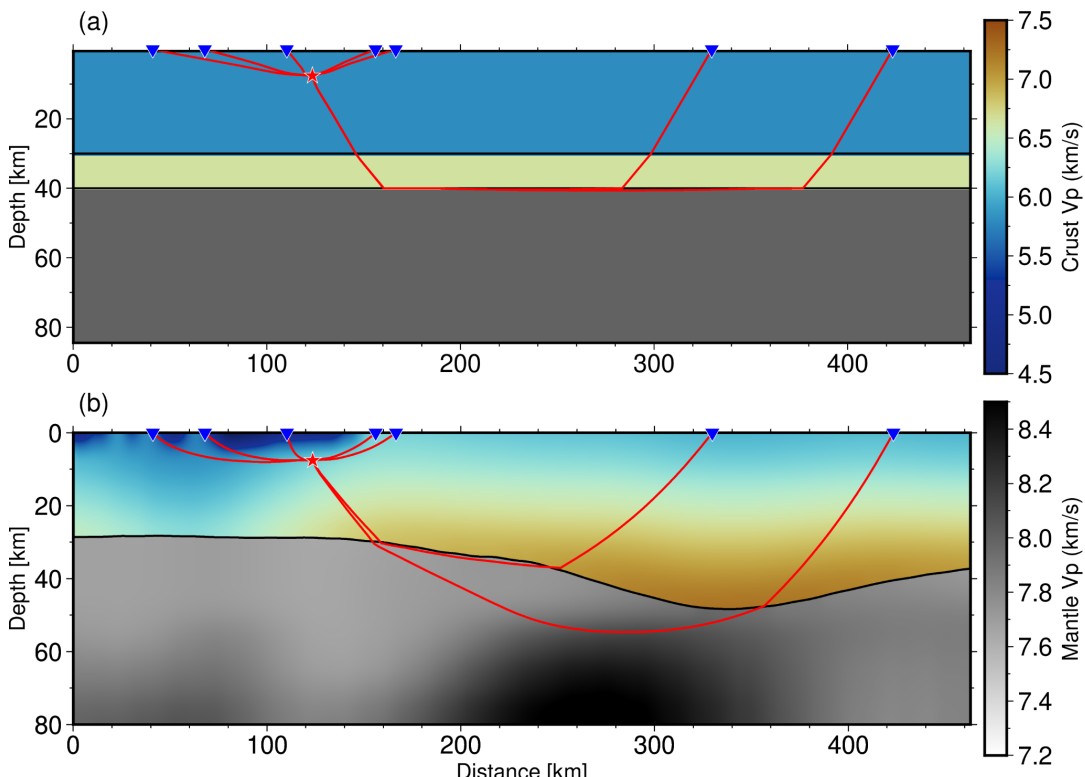


**Figure 12.** Earthquake ray paths in cross section BB' (for location see **Fig. 8**) for two models: (a) a simple 1D model with two isotropic crustal layers, and (b) our 3-D model with one anisotropic crustal layer. Colorbars are the same for both panels.

In addition to the profiles shown in **Figs. 11** and **12**, we have also extracted 1D depth velocity models for six points marked in **Fig. 8** in magenta. Those six points have been chosen to cover different domains of our model (Stable Adria, Internal and External Dinarides, and the SW Pannonian Basin). The velocity change with depth for the chosen six points is shown in **Fig. 13**. For example, at the P2 point, which is located in the SW Pannonian Basin, the velocity for the first

couple of kilometers of depth is much lower than for the other points because there is a Neogene deposits layer on top. The outlook of each model shown in **Fig. 13** is generally similar at each point with obvious differences being Moho depth (see points P2 and P1) and rate of increase of velocity with depth (e.g., compare points P3 and P5).



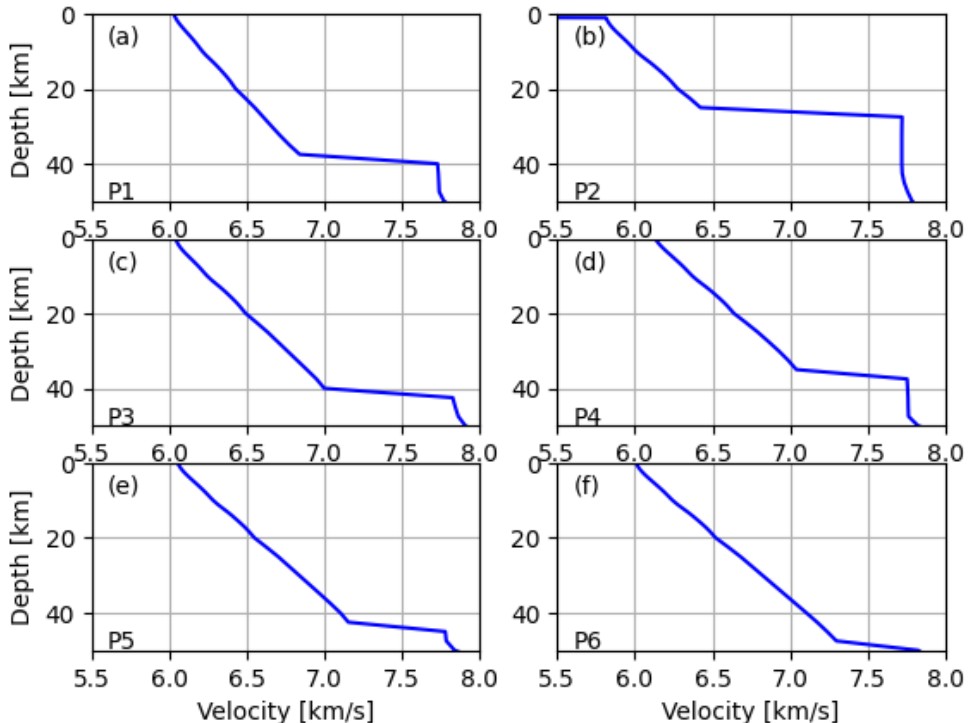

**Figure 13**. Velocity changes with depth for six points (P1 to P6; for locations see **Fig. 8**).

**Conclusions**

Having a complete 3-D model of the crustal structure is a major step forward in the study of the Dinarides and the surrounding areas. The newly derived model is defined on a regular dense grid for three key parameters (P- and S-velocity and density), and as such can be readily used by seismologists who need information on crustal structure as input for their studies (earthquake locating, seismic tomography, shaking estimation, seismic hazard…). We tested the performance of the model in comparison with the commonly used 1D model and found significant improvements in time travel calculations. The model still has some inherent weaknesses that have been discussed in the previous sections which are mostly connected with low number of measurements in some parts of the region. Nevertheless, the 3-D model represents a good first step towards improving the knowledge of the crustal structure in the complex area of the Dinarides.



The model clearly delimits several key areas (Dinarides, Pannonian basin and Adriatic Sea) as well as distinguishing distinct layers in that region (i.e., Neogene deposits and Carbonates). The most robust feature of the model is the depth to Mohorovičić discontinuity, but other parameters are also reasonably well defined. Inclusion of the Carbonate complex thickness is, to the best of our knowledge, the first attempt to estimate this parameter for the whole Dinarides region. One of the

new insights found during the creation of the model is the relatively high (P-wave) velocity for the lower crust in the southern part of the Dinarides. This feature needs to be confirmed by other studies, given the sparsity of information about velocity for that region. On the other hand, the high velocity feature fits nicely to the higher velocity of the uppermost mantle found in the same area thus corroborating the idea of continental subduction (and/or lithospheric delamination) in

south-central External Dinarides.

In conclusion, the model presented here represents the currently best and most complete crustal model for the wider Dinarides region. Model is assembled from all the available measurements on seismic velocity, density, layer composition and thickness to provide a full representation of the major variations of seismic wavespeeds in the regional crust. Hopefully, the new 3-D model will

help discover some new, previously unknown features of the crust and in turn, each new study that sheds some light on the crustal structure in this area may improve the 3-D model derived here.

**Resources**

The model derived in this work is available on the following link: https://urn.nsk.hr/urn:nbn:hr:217:793485

**Acknowledgements**

This work has been supported in part by the Croatian Science Foundation under the Project No. IP-2020-02-3960. This research was performed using the resources of computer cluster Isabella based in SRCE – University of Zagreb University Computing Centre. We thank professor Marijan Herak for the review and comments which helped us improve this manuscript.

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



## Appendix A

### Kriging

Kriging is a method of interpolation formulated for the estimation of a continuous spatial attribute (e.g., depth to Moho interface) at an unknown site, using the limited set of data from sampled sites. It is a form of generalized linear regression used for the formulation of an optimal estimator in a minimum mean square error sense (Olea, 1999).

Generally, the value at a point of interest is calculated as follows:

$$\hat{Z}(x_0) = \sum_{i=1}^{n} \lambda_i\, Z(x_i),$$

where $\hat{Z}(x_0)$ is value estimation at point of interest $x_o$, $\lambda_i$ are weights, and $Z(x_i)$ are known values at sites $x_i$. The kriging weights are derived from the covariance of the sampled values. The first step in kriging interpolation is estimation of the variogram (also called a semivariogram) – a statistic that assesses the average decrease in similarity between two random variables as the distance between them increases. It is the inverse of covariance – the covariance measures similarity, and the variogram measures dissimilarity. Unlike the other moments (e.g., the mean), the variogram is not a single number, but a continuous function of a variable $h$, called the lag. The variogram calculated from the sampled points is called the experimental variogram. The experimental variogram is not used in the calculation of kriging weights but is used to fit a theoretical variogram which in turn is used for calculation of the weights. When fitting, we can use limited types of semivariograms which have acceptable properties needed for solving the system of equations in order to obtain the weights. If we would use the experimental variogram directly, we might end up with an unsolvable system of equations. For example, **Fig. A1** shows an experimental and theoretical variogram used for interpolation of Moho interface depth. A variogram, such as the one in **Fig. A1**, that increases in dissimilarity with distance over short lags and then levels off is called a transitive variogram. The lag at which it reaches a constant value is called the range, and that constant value is called the sill. For the Moho depth interpolation, we had an abundance of data available, and we were able to estimate a theoretical variogram which fits the observed data almost perfectly.

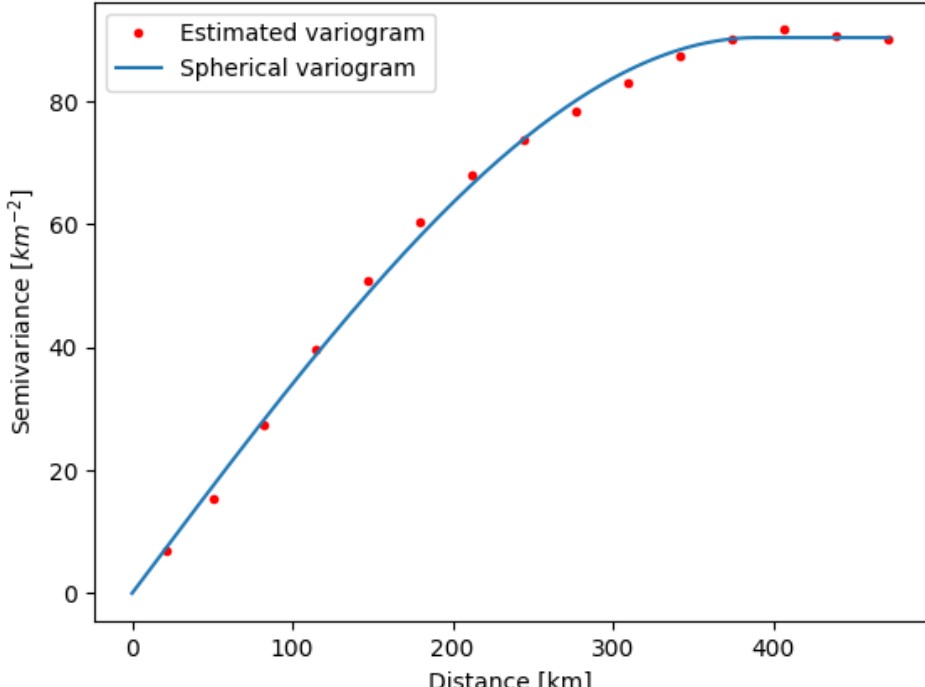

**Fig. A1**. Estimated and theoretical (spherical) variogram used for interpolation of Moho interface depth.

In case of other model parameters, we did not have such a perfect fit for larger lags. For instance,
**Fig. A2** shows the variogram used for Neogene deposits bottom interpolation. In this case, the
theoretical variogram was not spherical, but exponential. In this case, for the largest lags shown,
the theoretical and estimated variograms do not fit. For the calculation of the variogram pairs of
measured values are used. Since for the greater distances (greater lags) there are fewer numbers of
such pairs, the estimates are less accurate for those lags. Fortunately, for practical use in kriging,
the variogram closer to the origin requires the most accurate estimation (Olea, 1999), and we had
that condition met for all our model parameters.



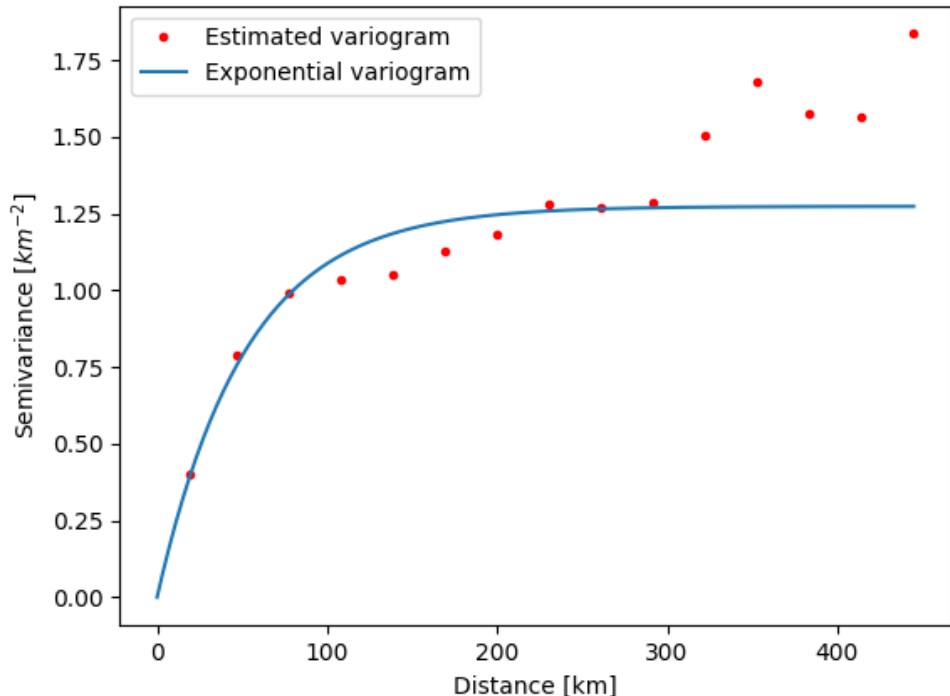

**Fig. A2.** Neogene deposits bottom estimated and theoretical (exponential) variograms.

Variograms for Carbonate complex bottom depth and velocity estimation are shown in **Figs. A3** and **A4,** respectively. Crust velocity variogram shown in **Fig. A4**. is required to have a constant

mean in the sample space to be able to estimate a variogram. In case of a gentle and systematic variation in the mean (called the drift), e.g., velocity increases with depth, it must be removed prior to the estimation of the variogram. Such a drift was indeed observed and was removed prior to the estimation of the variogram shown in **Fig. A4**.



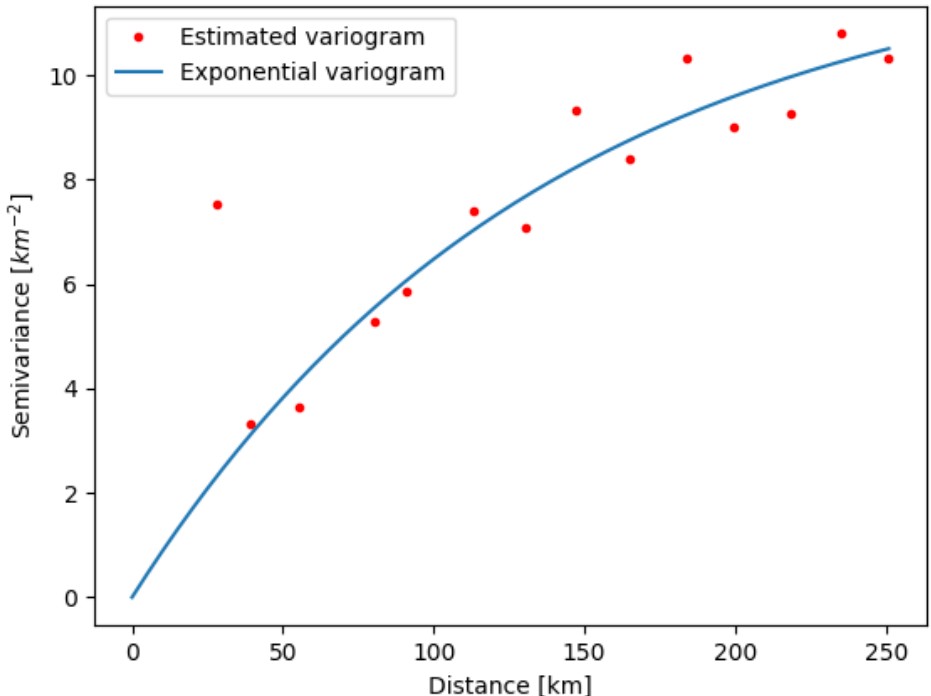

**Fig A3.** Carbonate complex bottom estimated and theoretical (exponential) variogram.



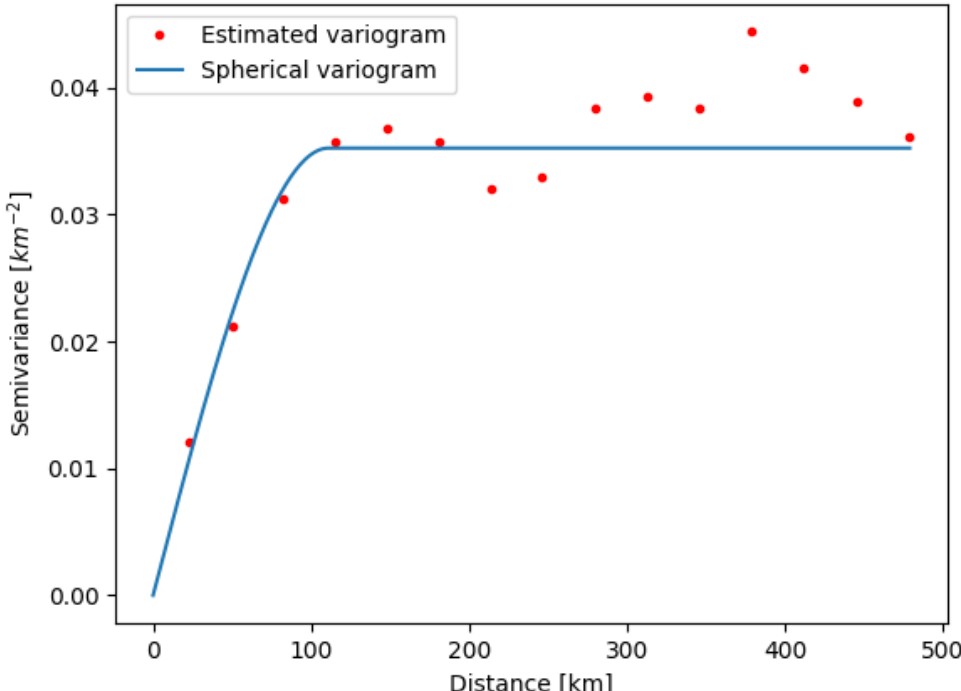

**Fig. A4.** Estimated and theoretical (spherical) variogram used for interpolation of the crustal velocity (Vp).

Once we had variograms estimated, we were able to obtain the weights and calculate the values of
parameters (Moho, Neogene deposits bottom, Carbonate complex bottom, velocity) for each point
in our grid. All the operations, both variogram estimation and the interpolation itself, were done
using the gstat package (Pebesma, 2004). Alongside the interpolated values, the package also
returns the variance estimates for each point in the grid.

The interface parameters (Moho, Neogene deposits bottom and Carbonate complex bottom depth)
were interpolated using ordinary kriging. Ordinary kriging assumes that the mean of the value is
unknown, but constant. For interpolation of the velocity, we had to use a more general type of
kriging – the universal kriging. It relaxes the condition on the mean – it is no longer assumed
constant. The other properties of kriging are shared between both types used. They are both
minimum square error estimates. The estimation is not limited to the data interval (it is possible to
extrapolate – although it is less accurate). They have, so called, declustering ability – the
measurements that are spatially clustered have lower weights than isolated points. They are exact
interpolators with zero kriging variance – meaning that if, for instance, we try to calculate the value
exactly at a sampled point, kriging will return the exact value and assign 0 variance to it. It can be
nicely seen in **Fig. 5** showing the velocity variances. Since the variance at sampled sites is zero it



is possible to discern the profiles that were sampled for data. Kriging is not able to handle duplicate
       points – it causes the insolvable systems of equations for the kriging weights – therefore we had
       to handle such points. It is also worth mentioning that the variance returned by the kriging software
       does not depend on variance or values of individual observations, but only on the sampling pattern.
       Therefore, we added the (estimated or available) data variance to the variance obtained from the
kriging and called it the total variance.