# Peer review of "Reference seismic crustal model of the Dinarides"

_EGUsphere, 2023_

## Referee Comment (RC1)

1) kriging of crustal parameters: did you observe any anisotropy in the variograms estimated along the vertical direction and along the horizontal axis?
2) how did you choose the width of the gaussian filter applied to the Moho interface?
3) It could be interesting to analyze how your model is different from European scale models (e.g. EPCrust).
4) Petrinja earthquake:
    a) which hypocenter did you used? You did not report the reference in the text.
    b) Fig 9 shows that the use of the 3D model removed the dependency of the residuals on the distance and greatly reduced the residual dispersion; on the other side the mean residual (for Pg and also Pn)ì) is about -0.5s: why did you not re-localize the event with the new velocity model?

- lines 255-266: not clear
- line 330: "we specified a relatively large area between 10° and 20° east longitude" -> from the maps (fig 3, …) it seems that the interpolated area reaches 20.5° E longitude
- lines 798-799: Handy 2010 is in the References but not cited in the text
- lines 813-815: Kennet et al 1995 is in the References but not cited
- lines 816-818: Korbar 2009 is in the References but not cited
- lines 886-888: Tari 1998 is in the References but not cited

---

## Referee Comment (RC2)

- lines 254 – 265: Please explain better, this part of text is unclear.

- lines 371 – 376: What velocities for Neogene deposits did you get with these relations? From-to??

- lines 409 and 410: Do you have the sides reversed (SE and NW)?

- Figure 4: Please edit the coordinate marks that overlap at the 20 and 30 km depth slices.

- line 571, 615: Brocher's

- lines 614, 615: Can you explain what relations you used? What is a similar age? The deposits mentioned range from the Lower Miocene to more recent times.

- line 618: you said earlier 6.2–6.3 km/s

- Figures 9 and 10: What hypocenter parameters did you use to calculate travel times? A hypocenter derived based on the mentioned 1D model or a new 3D model, or some other velocity model? It would be interesting to show the locations of the hypocenters in the studied area based on the commonly used 1D model and the new 3D model.

  **General:**
  o   Match the references in the text to the list.
  o   I recommend a careful proofreading.

---

## Referee Comment (RC3)

[referee-annotated manuscript omitted]

---

## Editor Comment (EC3)

Dear Editor,

Below please find explanations on comments from the reviewer and our replies. The reviewer accepted all our replies and was very positive about the revised manuscript. We accepted most of reviewers new technical corrections – see marked manuscript R2. Nevertheless, the reviewer left a couple of technical corrections to the Editors decision, and we would like to provide some explanations about those comments for your easier decision. The comment-replies are structured as specified below.

In red - reviewers initial comment
In black - authors initial reply
In green - reviewers final reply
(if needed) In black - authors additional explanation

**Comment #1**

It is not clear how useful it is to present maps showing S-wave velocity and density. In my opinion, they could be removed since they are the result of simple (empirical) P-wave velocity conversions. The resulting maps do not contribute anything to the main conclusion of this study.

Density was also calculated-interpolated, only the input data set was much smaller. We included the S-wave velocity for completeness (since we had P-wave velocity and density). This was done to facilitate easier usage of the S-velocity as a starting model for some future research (ambient noise or surface wave tomography, earthquake shaking estimation, etc.). If the reviewer feels that the S-wave model still needs to be removed, we can move S-wave figures to supplement.

The way the model parameters have been estimated is much clearer in this new version of the manuscript. I would leave the decision to the editor where to present the S-wave velocity model (main text or supplementary). As the description remains very short, it could stay in the main text as well.

We agree with the reviewer and would leave the S-model in the main text if the Editor agrees.

C.B.: I agree with leaving the Vs model as it may be useful for readers and for future use.

**Comment #2**

As an important point of the critical discussion of model limitations, the authors should comment on the fact that they have jointly interpolated P-wave velocities from active seismic profiles and those derived from gravity constrained density values. Is this reliable? There seems to be a systematic jump in velocity when crossing the boundary between the different domains. Does this mean that such abrupt changes in the interpolated maps do not reflect real differences in rock physical properties but inconsistencies due to different methods?

As the above mentioned data were the only available measurements in some areas there

wasn't much choice to begin with. How reliable some of the transitions are is hard to estimate but we strongly believe that a good part of "jumps" between different domains reflects true physical properties. The models acquired from gravity data which were used in our work were taken from work of Šumanovac (2010) where the author calibrated gravity modelling based on active seismic profile parallel to one of the gravity profiles. We have added the following paragraph to the discussion chapter:

"It seems that the velocity in the Internal Dinarides, where we only had inverted gravimetric profile data available, is slightly higher than in the rest of the model. At this point, we cannot discern if it is an actual feature, or some artefact due to lower quality data. The fact is that this is a different tectonic unit, so it is not impossible that it has different features. If we have omitted these data from interpolation, we would have even worse results, because in that case the values would be purely extrapolated. The approach we chose gave at least some constraint to the velocity values in this part of the model."

The authors have been more precise and more critical in describing the results in the revised manuscript. A new, general point on this chapter: the travel time calculations have been performed to test the presented 3D model. The tests appear in the "Discussion" chapter since the authors obviously want to discuss the reliability of the model in this way. However, since they present these tests with a description of the data / techniques used and results obtained, the "Discussion" chapter loses much of its "discussing" character. This is definitely not a standard way of structuring a scientific paper, but I would leave it to the editor at this very late stage of the review process, to decide whether the authors should modify this.

We agree with the reviewer that it's not a standard way of structuring but feel that leaving it within the discussion section greatly improves the readability of the manuscript and improves the overall discussion by including model testing within discussion on model characteristics.

C.B.: I agree with the reviewer and made a suggestion in the annotated PDF of the manuscript.

**Comment #3**

Line 434: Figure 3 - Choose the same colour palette for maps of the same physical unit. All interface depths (in m or km) including the Moho should have the same colour palette (here Moho depth is presented in the same colours as vp-values in Fig. 4, which should be avoided). Uncertainty (also in m, respectively km) could be shown by a different colour palette.

This was done on purpose. Notice that sediment and carbonate bottom depths have values going from 0 to a certain value (6 km and 15 km respectively), the chosen colour palette better shows the features (0 km is shown in white, while the maximum values are shown in black). Same is true for the errors. On the other hand, that colour palette does not adequately represent the features of the Moho, and therefore we chose the same one as for the Vp.

I would leave this decision to the editor, since this actuall regards the standards of the journal.

We stand by our first answer as this colour scheme greatly enhances visual representation of different values within the model.

C.B.: Here again I agree with the reviewer. Please, replot those figures in a more consistent manner. It will be easier for the readers if you employ a more "traditional" way of plotting your results.

**Comment #4**

Lines 620 - 622: To test how well the newly derived 3-D model represents the true structure, we calculated the travel times for a regional earthquake recorded on representative seismic stations in the wider Dinarides area (Fig. 8) - This model validation approach should be mentioned in the introduction, the description of the modelling approach and shortly also in the abstract.

In the abstract, we have rephrased the sentence: "The newly derived model has been compared with the simple 1D model used for routine earthquake location in Croatia, and it proved to be a significant improvement." to "To validate the newly derived model, we have calculated travel times for a regional earthquake recorded on a number of seismic stations in the Dinarides area. The calculated travel times have been compared with the travel times in the simple 1D model used for routine earthquake location in Croatia, and it proved to be a significant improvement." We haven't mentioned it in the introduction nor in the modelling approach to reduce repetition.

I appreciate that this important step is mentioned in the abstract now. Instead of describing the technique in the discussion chapter, however, I would move it to the "modelling approach" chapter (see also below).

On this one, we feel that it should stay in the "Discussion" section. The testing of the model does not have any connection with how the model was assembled i.e. "Modelling approach" and is much more useful when discussing the performance of the new model.

> C.B.: Please, see my comment above and in the annotated PDF of the paper regarding the discussion section

**Comment #5**

Lines 623 - 631: The 1D model's topmost layer is characterized by P-velocity of 5.8 km/s, and the deeper crustal layer has the P-wave velocity of 6.65 km/s. For the same model the uppermost mantle velocity is 8.0 km/s. We then compared the travel times from both models with the true measured travel times. We used the Pn and Pg phases of the 2020 Petrinja Mw6.4 earthquake. The location of the earthquake and the stations that recorded the wave onsets are shown in Fig. 8. For the same stations we calculated the travel times using the 1D and the new 3-D model. For travel time calculation we used the Fast Marching Method (de Kool et al., 2006) as implemented within the FMTOMO package (Rawlinson and Urvoy, 2006). - This is methodology and should be moved to a chapter "Modelling approach".

This is simply a description of the 1D model we used, and the means of calculating the travel times, so we think it should remain in this section.

I would still argue for moving this paragraph to the "modelling approach" chapter (see also the comment in the PDF), but I would leave the final decision to the editor.

As before we feel that describing this 1D model in the discussion gives better overall covering when testing the model. If we described this in the "Modeling approach" it would be

lost until we got to the point where we actually use this model. Also, we are not using this 1D model to create a new model but to test the performance of the new model hence it should be in the discussion.

C.B.: I agree with the reviewer.

**Comment #6**

Lines 656 - 657: generally closer to the actual observed travel times for all the epicentral distances shown than those calculated using the 1D model. - This is not obvious from the figure.

The abscissa in the figure shows the difference between observed travel time and travel time calculated from 1-D (black points) and 3-D (blue crosses) models. The differences with 3-D model generally lie closer to the dashed line, representing zero, meaning they are closer to the observed travel times hence the conclusion that generally travel times for 3D model are closer to the observed ones.

Can you still comment on why there is a trend of 3D model points to show negative time difference, while 1D model points mostly show positive values (for both Pg and Pn phases)?

The travel time calculated using the new model is slightly smaller than the observed travel times. Total travel time is influenced by all the velocity anomalies along the ray path, and at this point we cannot say for sure which part of our model is causing discrepancy. For Pn phases, the upper mantle is mostly affecting the travel time, but that one was not derived in this work, but simply taken from Belinić et al. (2020), and they have reported the potential problems (especially for the uppermost mantle which the Pn phases cross).

[revised manuscript text omitted]

---

## Author Comment (AC1)

Dear reviewer,

Thank you for your insightful comments that will improve our manuscript. Below are the answers (in black) to your comments (in red).

1) kriging of crustal parameters: did you observe any anisotropy in the variograms estimated along the vertical direction and along the horizontal axis?

There was no anisotropy observed. We observed a linear dependance on depth (the drift) in vertical direction, which has been taken into account during the variogram estimation. We added the following sentence in the Appendix explaining this:

*"Besides the drift, we estimated the experimental variogram in several directions, in order to check if it was dependent on the direction (i.e. if there was anisotropy), but we have not observed any anisotropy."*

2) how did you choose the width of the gaussian filter applied to the Moho interface?

We added the following explanation in the text:

*"The same smoothing was applied for the Moho interface and for the crustal parameters. We observed how it influenced the crustal velocity, particularly in the area of the model for which most data was provided by gravimetric profiles. Given that the gravimetric profiles are interpreted in terms of isotropic sections, and given that the smaller sections interpreted were roughly about 100 km in dimension along the profile, we chose this as the Gaussian width. It was also confirmed by trial and error that below this width we observe some artifacts in the model."*

3) It could be interesting to analyze how your model is different from European scale models (e.g. EPCrust).

We added an Appendix comparing some of the features from the regional EPCrust model and our new model.

4) Petrinja earthquake:
a) which hypocenter did you used? You did not report the reference in the text.
b) Fig 9 shows that the use of the 3D model removed the dependency of the residuals on the distance and greatly reduced the residual dispersion; on the other side the mean residual (for Pg and also Pn)ì) is about -0.5s: why did you not re-localize the event with the new velocity model?

(a) We used the Petrinja earthquake mainshock as it was well recorded on all the stations in the region. The location of this earthquake was 45.4188N, 16.2082E, 7.57 km; the information has been added in the text.

(b) We did not deem it necessary to relocate the earthquake, since it was the main shock of the series and was well recorded on a great number of stations. The point of this example was to show that travel times are greatly improved when using the 3D model. Nevertheless, we subsequently relocated the earthquake with the new model and the images below show the location after this. As expected the location stayed basically the same.

[Figure]

**Fig 1. a)** Mapview of the Petrinja earthquake used for testing the new model. Red start is the new location and black is the old. **b)** cross-section cutting through hypocenter location showing only minor depth variation between locations.

● lines 255-266: not clear

The explanation about how the thickness of the carbonate layer was acquired laid out in lines 255-266 was rewritten in a (hopefully) more clear and concise way. The new text is the following:

"*The interface with the least data at our disposal was the Carbonate rock complex (CRC) bottom depth. The CRC bottom depth was estimated combining geological and structural data published in available Basic Geological Maps at the 1:100,000 scale with accompanying Explanatory Notes that cover entire Dinaridic area, as well as geological-structural data published in studies of Tišljar et al. (2002), Vlahović et al., (2005) and Balling et al., (2021). Based on the collected data, we determined the spatial extent of the Paleozoic–Paleogene CRC. Since the CRC represents a very distinctive layer in the Dinarides, we additionally estimated its thickness. Assessment of CRC thickness was initially performed at the scale of each of more than 80 geological maps covering the study area, using thicknesses presented in geological columns on each map. Derived values of CRC thickness were further considered in respect to the deformation styles and large-scale structural relations (e.g., Balling et al., 2021). Several regional carbonate nappe systems in the External Dinarides characterized by extensive folding and thrusting could reach a combined stacking thicknesses up to 12000 m, but thicknesses are not evenly spatially distributed. Significant variability of the CRC total thickness in the Dinarides is caused by combination of (1) initial differences in thickness due to significant paleogeographic differences along the Adria Microplate passive margin, since a total thickness of the Adriatic Carbonate Platform and thick underlying and thin overlying carbonates is in the range of 4500–8000 m (Tišljar et al., 2002; Velić et al., 2002; Vlahović et al., 2005), (2) structural position of individual nappe systems in respect to the active collision front, and (3) variable strain rates and stress orientation during the Cretaceous–Paleogene Adria–Europe collision. Nappe stacking systems in the central and southern part of the External Dinarides, where CRC is the thickest (Fig. 3c), locally incorporate up to four thrust sheets composed of different segments of the entire carbonate succession.*"

● line 330: "we specified a relatively large area between 10° and 20° east longitude" -> from the maps (fig 3, ...) it seems that the interpolated area reaches 20.5° E longitude
This was a typo which we now corrected it in the text

- lines 798-799: Handy 2010 is in the References but not cited in the text
- lines 813-815: Kennet et al 1995 is in the References but not cited
- lines 816-818: Korbar 2009 is in the References but not cited
- lines 886-888: Tari 1998 is in the References but not cited

We apologize for leaving these references in the list. They were probably left from the first draft. The references which are listed but not cited in the main text are now removed.

---

## Author Comment (AC2)

Dear reviewer,

Thank you for your insightful comments that will improve our manuscript. Below are the answers (in black) to your comments (in red).

lines 254 – 265: Please explain better, this part of text is unclear.

The explanation about how the thickness of the carbonate layer was acquired laid out in lines 255-266 was rewritten in a more clear and concise way. The new text is the following:

"*The interface with the least data at our disposal was the Carbonate rock complex (CRC) bottom depth. The CRC bottom depth was estimated combining geological and structural data published in available Basic Geological Maps at the 1:100,000 scale with accompanying Explanatory Notes that cover entire Dinaridic area, as well as geological-structural data published in studies of Tišljar et al. (2002), Vlahović et al., (2005) and Balling et al., (2021). Based on the collected data, we determined the spatial extent of the Paleozoic–Paleogene CRC. Since the CRC represents a very distinctive layer in the Dinarides, we additionally estimated its thickness. Assessment of CRC thickness was initially performed at the scale of each of more than 80 geological maps covering the study area, using thicknesses presented in geological columns on each map. Derived values of CRC thickness were further considered in respect to the deformation styles and large-scale structural relations (e.g., Balling et al., 2021). Several regional carbonate nappe systems in the External Dinarides characterized by extensive folding and thrusting could reach a combined stacking thicknesses up to 12000 m, but thicknesses are not evenly spatially distributed. Significant variability of the CRC total thickness in the Dinarides is caused by combination of (1) initial differences in thickness due to significant paleogeographic differences along the Adria Microplate passive margin, since a total thickness of the Adriatic Carbonate Platform and thick underlying and thin overlying carbonates is in the range of 4500–8000 m (Tišljar et al., 2002; Velić et al., 2002; Vlahović et al., 2005), (2) structural position of individual nappe systems in respect to the active collision front, and (3) variable strain rates and stress orientation during the Cretaceous–Paleogene Adria–Europe collision. Nappe stacking systems in the central and southern part of the External Dinarides, where CRC is the thickest (Fig. 3c), locally incorporate up to four thrust sheets composed of different segments of the entire carbonate succession.*"

lines 371 – 376: What velocities for Neogene deposits did you get with these relations? Fromto??

The values we obtained from Brocher's (2008) relations ranged from around 0.7 km/s near the surface, to around 5.6 km/s at greatest depths of Neogene deposits (which was 7.5 km below surface). The information has been added in the text.

lines 409 and 410: Do you have the sides reversed (SE and NW)?

Indeed, we have made a mistake. It has been corrected. Thank you for pointing out this mistake.

Figure 4: Please edit the coordinate marks that overlap at the 20 and 30 km depth slices.
Figure 4 has been updated to correct this.

line 571, 615: Brocher's
Corrected in the text

lines 614, 615: Can you explain what relations you used? What is a similar age? The deposits mentioned range from the Lower Miocene to more recent times.
There are several velocity-depth relations derived in Brocher (2005) which are defined for different depth ranges. The relations derived for the shallowest depths are reported to be Plio-Quaternary, and for the greater depths the relations were derived using information from basins which were mostly deposited during the Miocene. As there is little information about these structures in the greater Dinarides area the best we could do was a first order approximation, and therefore using those relations seemed appropriate.

line 618: you said earlier 6.2–6.3 km/s
It has been corrected in the text, it must have been a leftover from an earlier version. Thank you for pointing it out.

Figures 9 and 10: What hypocenter parameters did you use to calculate travel times? A hypocenter derived based on the mentioned 1D model or a new 3D model, or some other velocity model? It would be interesting to show the locations of the hypocenters in the studied area based on the commonly used 1D model and the new 3D model.
We used the hypocenter from the Croatian catalogue (45.4188N, 16.2082E, 7.57 km). It was the mainshock of the recent Petrinja Mw6.4 earthquake which was well recorded on many stations, and its location is very well defined. We relocated it using the new model, but since it was so well recorded, there was not much difference between the two hypocenters (see Fig 1 below). The initial location was based on the 1D model mentioned in the main text with added station corrections. Relocations with the new 3D model were done for a large group of earthquakes from the main catalogue with mixed results, mostly connected with the setup of the Fast Marching Method (de Kool et al., 2006) as implemented within the FMTOMO package (Rawlinson and Urvoy, 2006). Whereas travel times showed significant improvement with the new 3D model relocations stayed close to the initial locations. As relocating proved to be a significant undertaking we opted to do this in connected paper (following this one) where we would do relocations in line with tomography and/or updating of a new model.

[Figure]

**Fig 1. a)** Mapview of the Petrinja earthquake used for testing the new model. Red start is the new location and black is the old. **b)** cross-section cutting through hypocenter location showing only minor depth variation between locations.

Match the references in the text to the list
There were some leftover references from the first draft, we have fixed the mismatch between the reference list and the text.

I recommend a careful proofreading

Done. Hopefully we minimised the number of mistakes in the text.

---

## Author Comment (AC3)

Dear reviewer, thank you for the insightful comments and time invested in reading the manuscript. Please see our answers (in black) to your comments (in red) below.

*Abstract*

The abstract does not ideally present the objectives, methodology, main results and implications of the study performed. This could be improved a lot to attract more readers.

We made changes to the Abstract which hopefully now addresses the comments raised by the reviewer.

*Introduction*

This chapter misses the clear statements of (i) which problem is at hand, (ii) why the authors are motivated to solve it (purpose of the 3D model!), (iii) which general strategy they have chosen to overcome the problem. Since the envisaged purpose of the model remains unclear in the current version of this chapter, it is impossible to assess and judge the quality of the modelling approach chosen.

In the Introduction our aim was to briefly reflect on the most important (seismic) crustal structure investigations in the Dinarides done so far. From this we draw the conclusion that there were numerous investigations of the various parts of the Dinarides but none has so far combined all these various results to create a full 3D seismic model. This would reflect on the points (i) and (ii). For the point (iii) we feel that it was already pointed out in the frame that we aim to collect the available results on the structure of the crust in the Dinarides to create a new 3D regional crustal model. Nevertheless, we reworked and rewrote the Introduction section to better emphasise all the points raised by the reviewer with special care taken to address number (ii).

*Tectonic and geological setting*

This chapter should be shortened, i.e. reduced to information that is relevant for the understanding of the modelling approach and later discussion of results. More detailed information about lithological characteristics of the model units (Neogene, Carbonate Complex, crystalline crust) would be required to assess the reliability of the rock physical properties of the final model.

Our view is that this section, in its current form, is necessary as it gives an overview of the tectonic-geodynamic processes that led to the complex crustal structure we see today. The aim is to introduce the reader to some of the peculiarities of the structural composition of the Dinarides and surrounding regions, such as thinning of the crust toward the Pannonian

basin or thick carbonate cover and existence of deeper alluvial sedimentary cover in some areas. In our opinion this section is not too long and gives relevant background information that links the technical part of the manuscript with some of the decisions made in the model construction.

Lithological characterization in such a complex area is too broad (even significantly shortened) to fit into this article and needed detailed comparison of our results with that information is heavily out of scope of this investigation and would not in that form contribute much to the overall goal of assembling a regional seismic model. Nevertheless, in the references provided in this section the lithological properties are widely discussed and interested readers can consult those publications if needed. Furthermore, for the published results of geophysical investigations we are using, the reliability of those results has been mostly tested and discussed.

*Data / Model construction*

The chapters "Data" and "Model construction" are not well structured (e.g., "Data" already includes information on the model construction). Furthermore, the original chapters simply fail in putting someone in the position to correctly reproduce the 3D model when using the same data - which is a primary requirement for a model to be reliable and acceptable. Therefore, for the sake of more clarity and less repetition in the text, I suggest to merge the two chapters into one called "Modelling approach" which would have the following structure:  First, describe and justify the intended differentiation of the model into four discrete model units (Neogene, Carbonate Complex, crystalline crust and mantle). Then, introduce the technicalities of the model building process (e.g., the interpolation method). Finally, present a sub-chapter for each model unit to refer to (i) the used input data sets, (ii) their processing (including equations, uncertainty assessment method) and (iii) all decisions taken by the authors including explanations (e.g., how to compensate for observational gaps; why and how to recalculate sediment thicknesses, etc.). For the later model evaluation process (Discussion), each model unit should already here come with a map showing the locations / distributions of all data that have been integrated into the modelling. Furthermore, this chapter should include all methodological aspects of the model validation process through travel time predictions (in a final sub-chapter).

In general, the authors should avoid to phrase interpretations in this results chapter.

In the submitted version of the manuscript, it remains unclear whether uncertainty has been assessed (calculated using equations – which?) or merely taken over from the original data sources.

Thank you for this excellent suggestion. We have now merged the two chapters as suggested and rearranged paragraphs to make them more clear. This new-merged chapter begins with the description of the data used, then we introduce kriging method, mention the need to estimate variances for handling of overlapping data, add some more explanation on the total error estimate of the model (as combination of input data error and error from interpolation itself), and then describe in detail how the results were acquired from the available data. The last paragraph of the new (merged) chapter describes how we smoothed the final model.

*The main characteristics of the model*

It is not clear how useful it is to present maps showing S-wave velocity and density. In my opinion, they could be removed since they are the result of simple (empirical) P-wave velocity conversions. The resulting maps do not contribute anything to the main conclusion of this study.

Density was also calculated-interpolated, only the input data set was much smaller. We included the S-wave velocity for completeness (since we had P-wave velocity and density). This was done to facilitate easier usage of the S-velocity as a starting model for some future research (ambient noise or surface wave tomography, earthquake shaking estimation, etc.). If the reviewer feels that the S-wave model still needs to be removed, we can move S-wave figures to supplement..

*Discussion*

As an important point of the critical discussion of model limitations, the authors should comment on the fact that they have jointly interpolated P-wave velocities from active seismic profiles and those derived from gravity constrained density values. Is this reliable? There seems to be a systematic jump in velocity when crossing the boundary between the different domains. Does this mean that such abrupt changes in the interpolated maps do not reflect real differences in rock physical properties but inconsistencies due to different methods?

As the above mentioned data were the only available measurements in some areas there wasn't much choice to begin with. How reliable some of the transitions are is hard to estimate but we strongly believe that a good part of "jumps" between different domains reflects true physical properties. The models acquired from gravity data which were used in our work were taken from work of Šumanovac (2010) where the author calibrated gravity modelling based on active seismic profile parallel to one of the gravity profiles. We have added the following paragraph to the discussion chapter:
"*It seems that the velocity in the Internal Dinarides, where we only had inverted gravimetric*

*profile data available, is slightly higher than in the rest of the model. At this point, we cannot discern if it is an actual feature, or some artefact due to lower quality data. The fact is that this is a different tectonic unit, so it is not impossible that it has different features. If we have omitted these data from interpolation, we would have even worse results, because in that case the values would be purely extrapolated. The approach we chose gave at least some constraint to the velocity values in this part of the model."*

**Technical corrections**

General: check and synchronize the order of figures in the paper and in the text

This was resolved.

Line 33: refer already here to the map of Figure 1 to introduce the location of the study area

This was resolved after the comment at line 144 has been resolved.

Figure 1: add the exact position of all active seismic profiles as this is the most important type of data for the study

The positions of the seismic profiles shown are their exact positions. We have slightly rephrased the Figure 1 caption to point out that these are actual positions: "*Blue lines mark the positions of Alp01 and Alp02 profiles (Brückl et al., 2007) – only the full line parts are used in the study; the red line is the position the Alp07 profile (Šumanovac et al., 2009); black lines are the positions of gravimetric profiles GP-1 to GP-6 (Šumanovac, 2010). (...)*"

Please consider also the detailed comments and suggestions in the uploaded PDF.
Line 1: rephrase to "reference"

Thank you. Rephrased.

Line 10: Abstract - Not informative, main message missing. Define "referent" seismic crustal model

Referent is used in the sense that it is the first such seismic model for the Dinarides, constructed using all the currently available results and that it will be freely available. Hence our aim is to make it a "to go" model for all future usage and that it will be refined (by us and  the community) as more data becomes available.

Line 13: overlain - check language

This sentence has been rephrased to: "*The good example of this are the Dinarides where thick carbonates cover older crystalline basement units and remnants of subducted oceanic crust.*"

Line 16: analysis - of what?

This has been rephrased to "*any seismic or geological analysis*"

Line 17: less - than what

Rephrased "less data coverage" to "*lack of information on crustal structure*"

Line 18: complete - Can a model be complete?

Rephrased to "*comprehensive*"

Line 21: seismic velocities (P- and S-) - How acquired? As tomography or profiles?

Neither. Seismic velocities in the model are acquired by kriging interpolation. We have added a sentence before this one: "*We have used kriging interpolation to obtain the model parameters.*"

Line 30: all the forthcoming studies - not clear? Examples?

The sentence has been rephrased as follows: *We hope that the newly assembled model will be useful for the forthcoming studies (e.g. as a starting model for seismic tomography, underlying model for earthquake simulations) which require knowledge of the crustal structure.*

Line 32: wider Dinarides region - refer to Fig. for location

The wider Dinarides region here is a pretty loose term describing Dinarides proper and parts of adjacent areas such a SW Pannonian basin, Eastern Adriatic Sea, Southern Alps. After moving a paragraph from Line 144 here, it is more clear what wider Dinarides region is. Reference to Figure 1 is also present in the moved paragraph.

Line 46: virtually - meaning?

That is the exact choice of words reported in the studies by Šumanovac, 2010 and Šumanovac et al., 2016 which reported one layered crust in the Pannonian basin. We agree that the term is not exact so we removed it.

Line 78 - 79: "Even though our crustal model is focused on the Dinarides, part of it also covers the SW margin of the Pannonian basin and Adriatic Sea." - refer to map

We added reference to Figure 1.

Line 121: 8000 m - add "of thickness"

Thank you for the suggestion, it has been added.

Line 144: The main objective of this study - Move this paragraph to the "Introduction". What is meant by "referent"? Explain the purpose of the model and give examples for later usage.

The paragraph has been moved to the beginning of the "Introduction" section and reworked to fit better in the Introduction.
What is meant by "referent" has already been answered at comment on line 10.

Lines 150 - 151: Our basic approach was to create a one-layered crust with laterally and vertically variable parameters (seismic velocities and density). - Rephrase: "...approach initially was...". This raises questions since the motivation for building the model is not well explained: - why seismic velocities and density?; - why a one-layered crust?

Thank you for pointing this out, we have modified the sentence, and now it is: "*We have decided to create a one-layered crust with laterally and vertically variable parameters.*" We have also added additional two sentences to explain why we chose to create a one-layered crust with the specific parameters: "*The reason for a choice of one-layered crust was a fact that not all the input data had the interpretation of intra-crustal interfaces, and those that had, did the interpretation differently. We used seismic velocities and density as parameters, since that is the data we had available.*"

Lines 161 - 161: Here, velocity and density of the carbonate layer were obtained using the same velocity (and density) data we had available for the rest of the crust. - This is not clear. How can these parameters be "obtained" if there are "no available data" (as stated in the previous sentence)?

Indeed, this is not very clear. Thank you for pointing it out. We have rephrased the sentence: "*The velocity and density of the carbonate complex rocks were not interpolated separately, but were interpolated using the same input velocity (and density) data we had available for the rest of the crust.*"

Line 176: Figure 1. - This map should present the exact location of active seismic profiles since these are the most important and reliable source of information on vp-values.

This map is already presenting the exact locations of the profiles, and the exact extent of the other data used. We have slightly rephrased the Figure 1 caption to point out that these are actual positions: "*Blue lines mark the positions of Alp01 and Alp02 profiles (Brückl et al., 2007) – only the full line parts are used in the study; the red line is the position the Alp07*

*profile (Šumanovac et al., 2009); black lines are the positions of gravimetric profiles GP-1 to GP-6 (Šumanovac, 2010). (...)"*

Table 1 header: obtained - replace by "processed"

Thank you for the suggestion. Replaced.

Line 191: three - plus the topography/bathymetry

That is correct. The sentence has been rephrased as follows: *"There are four interfaces defined in the presented model – CRC bottom, Neogene deposits bottom, Mohorovičić discontinuity (Moho), and topography/bathymetry."*

Lines 195 - 198: Since there was no velocity nor density information readily available for the Carbonate layer, its parameters were not assessed separately, as was the case for the Neogene deposits layer, but were calculated the same way as in the rest of the crust. The Carbonate bottom interface was kept in the model for the sake of completeness. - Repetitive (see above). Still the reasoning is not clear.

Removed this part from the text.

Line 209: no detailed error estimates - Unlogic: further below, the authors write "That estimate nicely fits with the error estimates reported by Bruckl et al. (2007) and Šumanovac et al. (2009) for refraction and wide-angle reflection profiles used in this study."

The point here is that there were not error estimates per-grid point like in other studies, but they were given generally, for the entire profile (±2–3 km, as mentioned a few sentences above). Error estimate of 6% of Moho depth (for each point we had!) indeed corresponds to the interval of ±2–3 km reported for profiles (generally!).

Lines 211 - 212: include the error estimates - How are these estimates "included"?

As suggested by the reviewer we merged "Data" and "Model construction" chapters and the text is rearranged in such a way that the first mention of the total error is a combination of errors from input data and errors from interpolation itself. We hope this made the statement more clear.

Line 213: Moho model - for the European continent

Thank you for your suggestion, it has been added in the text

Line 223: depth - For clarity add "base" depth.

Thank you for your suggestion, it has been added in the text

Lines 223 - 224: The most important region of the study area - unlogic, better: "Largest volumes of ... are situated..."

Thank you for your suggestion, it has been corrected in the text.

Line 224: loose - better "unconsolidated"

Thank you for your suggestion, we've made the correction in the text.

Lines 233 - 234: We did not use interpolation in this step, in order not to introduce an additional interpolation error. - Not clear. The topic of interpolation has not been raised until this point in the manuscript. It is confusing then to read here that it is NOT used.

Here we tried to point out that we used the data in the same form as was digitised in - every 5 km along the isolines, we did not interpolate it before adding it to other data. But you are right about mentioning interpolation at this point, the sentence added a bit of confusion so we removed it from the text.

Line 234: the error estimation - Again, the authors mention an error estimation the procedure of which is not introduced and explained.

This has been improved by rearranging "Data" and "Model construction" chapters. Please refer to replies in the beginning (after the comment on the chapters) and Line 211.

Lines 235 - 236: Since there was no error estimate beforehand, we decided to use the same percentage for Neogene deposits depth estimate. - This is not logic. Modelling errors generally arise from observational gaps, which typically increase with increasing depth. Assuming that the Moho depth errors of a continental scale model are applicable to a regional Moho may still be understandable, but applying them to a much shallower interface as well is very questionable.

As we stressed numerous times throughout the manuscript the data used come from a variety of sources some of which did not provide any error estimates. In this particular case for the Neogene deposits base depth data was taken from Saftić et al. (2003) and Matenco and Radivojević (2012) and there were no error estimates provided. We obtained the Neogene deposits depths from georeferenced isolines by digitising the maps provided in both studies. In order to provide at least some indication of error estimate we used a similar approach as was reported by Grad et al. (2009) where the authors for similar dataset and similar approach of manually digitising isolines estimate 15% error. Yes, we are aware that this is not an ideal solution but we deem that at least some error estimate is better than none in the case where the authors did not provide any error estimates or any digital data but only printed isolines. In this sense we strongly believe that our approach is logical and contributes to overall result and kriging interpolation stability.

Modelling (interpolation) uncertainties from Kriging are of-course calculated and reported as can be seen from Figures 3. and 4. in the manuscript.

Lines 238 - 239: since the basement is overlain by a thick layer of Mesozoic Carbonate rock complex. - This is not a valid explanation for missing Neogene strata. Rephrase to avoid a causal relationship (avoid "since").

Thank you for pointing this out, we have rephrased the sentence: "*The area of the Dinarides does not contain a Neogene soft deposits cover, at least not of significant thickness. The basement is overlain…*"
Of-course there are numerous smaller intrakarst basins filled with Neogene deposits but the data about these is scant and not easy to use. Furthermore, this data can easily be added in the future versions of this model and we encourage users to do so for smaller case studies.

Line 245: The interface with the least data at our disposal was the Carbonate complex bottom depth. - This compares data availability for different interfaces of the model. This point requires, however, a series of maps showing input data points for each modelled unit separately.

Thank you for your suggestion, such a map has been included in the supplement, and an additional sentence has been added before Fig. 1: "*The exact locations of data points used are shown in Appendix B.*"

Line 247: Basic Geological Maps - Why are there no geological maps mentioned in Table 1?

Thank you for your suggestion, they have been added to Table 1.

Line 251 - 252: Since the Carbonate complex represent a very distinctive layer in the Dinarides, we additionally estimated its thickness. - Reasoning of the sentence questionable. The actual reason for estimating the thickness of this unit should be the availability of thickness information (next sentence). Since the authors want to integrate all available information, this step is legitimized.

Thank you for pointing this out, we have left out that sentence.

Line 255: recalculated - This is a critical point.

- Why were thicknesses recalculated? If their present-day thickness is observed, this would be unlogic for a model that should represent present-day configurations (paleo-thicknesses won't be useful here)

- How have these recalculations been performed? The manuscript lacks a description that would allow a reproduction of the model with the given data.

Thank you for pointing this out. The entire paragraph was rephrased to make it more clear: "*Derived values of CRC thickness were further considered in respect to the deformation styles and large-scale structural relations (e.g., Balling et al., 2021). Several regional carbonate nappe systems in the External Dinarides characterized by extensive folding and thrusting could reach a combined stacking thicknesses up to 12000 m, but thicknesses are not evenly spatially distributed. Significant variability of the CRC total thickness in the Dinarides is caused by combination of (1) initial differences in thickness due to significant paleogeographic differences along the Adria Microplate passive margin, since a total thickness of the Adriatic Carbonate Platform and thick underlying and thin overlying carbonates is in the range of 4500–8000 m (Tišljar et al., 2002; Velić et al., 2002; Vlahović et al., 2005), (2) structural position of individual nappe systems in respect to the active collision front, and (3) variable strain rates and stress orientation during the Cretaceous–Paleogene Adria–Europe collision. Nappe stacking systems in the central and southern part of the External Dinarides, where CRC is the thickest (**Fig. 3c**), locally incorporate up to four thrust sheets composed of different segments of the entire carbonate succession.*"

Line 266: Fig. 3c - Refer to figures in their consecutive order! Either move this Figure up or refer to it in the main text just after Fig. 2.

This has been fixed by rearranging "Data" and "Model construction" chapters.

Line 281: It seemed the most logical course of action, assigning the density value to the middle of the layer, to represent the entire layer. - This is a point for discussion and it is written in the style of a discussion. Hence move it there.

This merely explains how we digitised data from gravimetric profiles, and we think it should stay in the section that deals with data preparation.

Lines 293 - 294: we assigned the maximum error estimate (to make a conservative estimate) - Estimating errors of interface depths obtained by gravity modelling is a very complicated task because of the general non-uniqueness of gravity interpretation. Such errors mainly depend on gravity field data errors and the amounts and distribution of gravity-independent data integrated. For this reason, it does not make any sense to transfer error estimates from one gravity derived model to another. I suggest not to give any error in this case.

This is not an error estimate of interface depth, but of density value. We thought it reasonable to give a maximum error estimate from the NAC model, because authors have used the same sources for their density values.

Line 314: 3 arc minutes in our model - This is an important characteristic of the model and the information should be given outside of brackets. Rephrase the sentence: a "regridding" has been performed.

This is merely a comment on the difference between the gridding of the SRTM15+V2.0 model and the model we derived here. The grid size of our model is mentioned later in the text. The sentence has been rephrased according to reviewers suggestion.

Line 336: 13° E and 20° E, and 42° N and 47° N. - Mention which extent has been used to present the results (maps below). Maps (below) show a slightly different coordinate range - why?

The range reported is the range we consider more reliable (hence the shaded areas in the figures). The maps are shown with a somewhat wider coordinate range, which we used for interpolation, and later marked as less reliable based on estimated errors.

Line 371: Brocher (2008) - Please provide the relevant equations.

We omitted the equations because it would be merely copying them from the original article. We have added the following to the sentence "*(eqs. 1, 3, 7, and 9 in the original article)*", so the interested reader is pointed to equation numbers in the referenced article. If the reviewer insists on listing actual equations, we will add them in the supplement but we will just be cluttering the already complicated procedure.

Lines 378 - 381: After collecting and preparing the data as described in the previous sections, the next step was interpolation. Ordinary kriging was used to interpolate model interfaces and universal kriging when interpolating the layer properties (P-velocity, density) as the layer properties are distinctly linearly dependent on depth. - This repeats information given before.

This has been fixed by rearranging "Data" and "Model construction" chapters.

Line 381: smoothed Moho discontinuity - Say how to guarantee model reproducibility. Say why: what makes a smoothed Moho advantageous compared to a non-smoothed version?

This sentence has been rephrased: "*After interpolation, we filtered Moho discontinuity and layer properties with a 100-km wide Gaussian filter to smooth the transitions between different data sources. The smoothing in case of the Neogene deposits and CRC interfaces was omitted because the data used in derivation of those interfaces came from similar sources,...*"

Lines 390 - 393: Neogene deposits and Carbonate bottom depths have not been smoothed, since they have been assembled from a small number of equivalent sources. Moho discontinuity, on the other hand, is assembled from a variety of different sources, and

therefore was smoothed using a Gaussian filter with a 100 km width. - Avoid repetition. Information belongs to "modelling approach" chapter.

These two sentences have been removed.

Line 398: Sava and Drava depressions - Since the locations of these depressions are not shown in the maps, nor is any other information about these structures given, referring to them in the text is not helpful.

Thank you for pointing this out. We have created a less cluttered map, which now shows the locations of the said depressions.

Line 401: That was expected, since - This is in the style of a discussion which should be avoided in the "results" part of the paper. Why was that expected? If there is a geological/tectonic reasoning that confirms the mere mathematical interpolation results, this would be a contribution to the discussion of the model quality.

We agree. The form of the sentence is somewhat imprecise and has been rephrased: "*In the SW Pannonian Basin, where Neogene deposits are the thickest, the underlying carbonate layer is thin, and vice versa…*"

Line 408: As corroborated before - Style. Please avoid repetition.

That sentence has been removed.

Line 417: Fig. 5 - Check figure order. Make sure you refer to Fig. 4 before referring to Fig. 5. In case, rearrange figures to fit the line of arguments.

This sentence has been removed. The reason for referring to Fig. 5 in this particular place was to point out the mentioned feature is better seen in it.

Lines 422 - 423: one can distinctly see the areas with less data coverage as areas with higher uncertainty. - This is indeed very difficult to see! Figure 2 shows data coverage for all interfaces and properties in one map. However, the areas of larger uncertainty (shadowed in maps) should be different for the different units / properties. How was the shadowed area defined, what does it actually mean? This is not clearly explained.

We tried to estimate the areas in which the model was more accurate, and to make such an estimate, we looked at total errors for Moho and for P-wave velocity (which were the best constrained parameters we had). This is mentioned in the new "Modelling approach" chapter.

Line 432: measurements - Put into a map all locations at which thickness was measured.

Such a map has been added in the supplement. Please also see the reply to the comment at line 245.

Line 434: Figure 3 - Choose the same colour palette for maps of the same physical unit. All interface depths (in m or km) including the Moho should have the same colour palette (here Moho depth is presented in the same colours as vp-values in Fig. 4, which should be avoided). Uncertainty (also in m, respectively km) could be shown by a different colour palette.

This was done on purpose. Notice that sediment and carbonate bottom depths have values going from 0 to a certain value (6 km and 15 km respectively), the chosen colour palette better shows the features (0 km is shown in white, while the maximum values are shown in black). Same is true for the errors. On the other hand, that colour palette does not adequately represent the features of the Moho, and therefore we chose the same one as for the Vp.

Line 443: separate - meaning seismic independent constraints? Or what?

Indeed, that is what it means. We have rephrased the sentence: "*Given that we had no independent estimate for the velocity for the CRC we cannot discern it as a separate layer just looking at velocity values.*"

Line 444: In the rest of the model - meaning in the sub-sediemantary crystalline crust?

No, meaning outside of the External Dinarides. Map in Fig. 1 has been refined to better show the geological features mentioned in text, and the sentence has been rephrased to be "*Outside the External Dinarides, one can see…*"

Lines 449 - 452: It is hard to discern if this reflects the actual structure, or if it is the consequence of the higher uncertainty in that part of the model (the velocity here was estimated from the density values from the gravimetric profiles, given that there were no other data sources available). - This is an important point for the discussion: it is highly questionable if one should merge at all seismic velocity values from active seismic experiments with those derived from gravity-constrained density values. This may produce method-related variations not representing the actual situation. Calibrations points would be required, i.e. locations where both types of velocity values are available.

The values for seismic velocity and density derived from gravity were taken from work of Šumanovac (2010) where the author calibrated the inversion of the gravity data based on the results from active seismic profile close to and subparallel to one of the gravity profiles. For details about calibration of seismic and gravity data please see that study.

Line 460: here - where? unclear!

Indeed, it is unclear. We meant Figs. 4 and 5. It has been corrected in the text.

Line 463 - 464: using the standard P over S-velocity ratio for the upper mantle. - This requires an equation and the corresponding citation.

The sentence has been rephrased to: "*...using the standard P over S-velocity ratio for the upper mantle (Vp/Vs = 1.73).*"

Lines 467 - 468: and in areas where data from active seismic profiles were available (Brückl et al., 2007; Šumanovac et al., 2009) - please show the exact locations of active seismic profiles in Figure 1 for the reader to assess the relationship with the presented uncertainty.

The locations of profiles shown in Fig. 1 are their actual locations. Please see also the comment for line 176.

Line 474: Figure 4 - Abrupt changes in uncertainty (e.g. between the NAC model area and the gravity-constrained area) are visible as abrupt changes in seismic velocity. This points to systematically biased velocity distributions as a result of an unreliable merging of data.

The major contribution to the total error shown in figure 4. is the error from interpolation itself, and the errors shown in figure are expected, since we knew the amount of data we got from each of the models. As far as the velocity values go, it is the best we could do. If we did not include gravity data, we would have an area without any input data, and it would result in much larger errors in that part of the model, and even worse situation with velocity values. We are well aware that gravity data is much inferior compared to the other data sources we used, but simply had no other data to use.

Lines 474 - 477: Velocity model depth slices and corresponding uncertainties: (a) model at a depth of 5 km, (b) uncertainty at a depth of 5 km, (c) model at a depth of 10 km, (d) uncertainty at a depth of 10 km, (e) model at a depth of 20 km, (f) uncertainty at a depth of 20 km, (g) model at a depth of 30 km, and (h) uncertainty at a depth of 30 km. - Could be shortened by less repetition. For help, just check other papers with figure panels.

Redone. Thank you for suggesting this.
"*Velocity model depth slices and corresponding uncertainties for 5 km, 10 km, 20 km, and 30 km depth are shown in panels (a)(b), (c)(d), (e)(f) and (g)(h). In the panel (c) the positions of the profiles shown in Fig. 5 are marked. The areas of lower resolution are shaded, and the grey colour scale corresponds to the mantle velocity (see text for details).*"

Lines 477 - 478: The areas of lower resolution are shaded - How are they defined?

This was mentioned in the new "Modelling approach" chapter. Please see the reply to the comment for "Modelling approach" section and Line 422.

Line 480: Fig. 5 - Why not showing profiles that correspond in location to the gravity profiles used as input data? What is the reasoning behind the choice of the six profiles?

We tried to pick profiles which cover the Dinarides perpendicular to their axis, because that is where the most interesting features can be seen. Moreover, these profiles are actually close to the locations of the gravity profiles and can be easily compared. Also, we chose one profile along the Dinarides' axis, to show how the features change along strike.

Line 480: b - c

In one of the working versions of text, this was indeed panel b, we haven't noticed that we didn't make the change in all places it was mentioned. The error has been corrected in the text.

Line 490: almost perfectly - rephrase to "in large parts of the profile"

Thank you for your suggestion, we've made the correction in the text.

Line 490: value - rephrase to "isoline"

Thank you for your suggestion, we've made the correction in the text.

Lines 515 - 516: This can be linked with the remnants of the subducted lithosphere and the ongoing underthrusting or lithospheric delamination (see e.g., discussion in Stipčević et al. 2020) - This is interpretation to be removed from this "results" chapter!

This sentence has been removed, because it carries the same information as the one in comment for Line 536.

Line 529: Figure 5 - The colour for lowest velocity of the crust is actually black due to the tightness of isolines. Therefore, these areas cannot be differentiated from the black regions of highest velocity in the mantle. The figure is not unambiguous and requires respective modifications.

Thank you for your suggestion, a modified figure has been added in the text.

Line 529: shown - rephrase to "with locations shown"

Thank you for your suggestion, we've made the correction in the text.

Line 529: b - c?

The error has been corrected in the text.

Line 529: AA' - "A" and "A'" should be shown at the ends of the profiles in the figure!

Thank you for your suggestion, a modified figure has been added in the text.

Lines 529 - 530: Parallel full lines running approximately along the profile are the velocity gradient lines. - Rephrase the sentence, remove approximately. These are isolines of velocity.

Thank you for your suggestion, we've made the correction in the text.

Lines 536 - 544: These positive anomalies in the work of Belinić et al. (2020) have been interpreted as a signal of the subducting Adria Microplate. Our model is mere interpolation of what was already known, but perhaps what we see here is part of the Adria crust being dragged along the uppermost part of the mantle being subducted below the Dinarides. As can be seen in the Fig. 5a, which is crossing into the SW Pannonian basin, the crustal velocity in that part is much lower than in the Dinarides, a feature also observed by Šumanovac et al. (2009). Perhaps the lower velocity is a feature of the Pannonian crust, whereas the relatively higher crustal velocity is a feature of the Adria crust. To make a definitive conclusion, more investigation should be performed. - Move to Discussion chapter!

Thank you for your suggestion, this whole paragraph has been moved to the "Discussion" chapter.

Line 559: The areas of lower resolution are shaded, and the gray color scale corresponds to the mantle density (see text for details) - It is hardly possible to discriminate between shaded and gray areas in the map. Suggestion: replace shading through hatched overprint.

Thank you for the suggestion, we have improved this figure with different shading. We have also updated the figures representing the other parameters.

Line 569: S-wave velocity at four depths. - As it is a mere linear conversion from vp, does it provide any new and relevant information?

We included the S-wave velocity for completeness (since we had P-wave velocity and density). This was done to facilitate easier usage of the S-velocity as a starting model for some future research (ambient noise or surface wave tomography, earthquake shaking estimation, etc.).

Lines 580 - 581: the need for the more complete seismic model of the Dinarides - From what does this need arise? Who would need this model and for which purpose(s) could it be used?

The knowledge of crustal structure is needed in any seismological study that deals with waves propagation (tomography, earthquake location, source inversion, etc.). One could

easily state that having a good starting (referent) model is of utmost importance in any seismological study. Moreover, the existence of a comprehensive model readily available in digital form greatly simplifies the preparation of input data for any such study.

Line 585: enough - What makes it be "enough"?

It never is "enough". It has been rephrased to "*most*".

Lines 585 - 586: It is mainly based on Stipčević et al. (2020) study and our model reflect all the features noted therein. - This is repetitive and trivial.

The sentence has been removed.

Lines 587 - 591: As shown in Fig. 5f the Moho deepens going from the NW to the SE along the main axis of the Dinarides. Also, from the profiles in Figs. 5a–e the change in Moho depth going from the Adriatic Sea towards the Dinarides mountain chain is much more gradual than on the other side, going from the Dinarides towards the SW Pannonian basin, where this change is much more abrupt, almost step-like. - This is a repetition of results, while here the reliability and relevance of these outcomes should be discussed.

The sentences have been replaced with: "*It confirms what we already know about Moho in the Dinarides, but now it is presented as a comprehensive, ready-to-use model.*"

Line 591: The same feature has been observed by Šumanovac et al. (2009) and Šumanovac (2010). - So, does the model provide any new results at all? What would these new results be?

The model has been assembled from existing results; it cannot provide "new" results, but now it is presented as a comprehensive, ready-to-use model. The sentence has been removed.

Lines 592 - 593: In the case of Neogene deposits thickness the data used came from two sources (Saftić et al., 2003 and Matenco and Radivojević, 2012) - repetitive

The sentence has been removed.

Lines 595 - 596: this parameter was adequately presented in our model. - What is the evidence for this statement?

We agree that this statement was a bit subjective. Therefore we rephrased it to the following:
"*For the Neogene deposit thickness we used manually digitized maps, therefore having less precise data, but which were originally created from a high number of active seismic profiles*

*and thus strengthening our confidence that this parameter was adequately presented in our model."*

Lines 596 - 597: Sava and Drava depressions - Again, without showing these structures, readers that are not familiar with the region will not understand this.

Map in Fig. 1 has been refined to better show the geological features mentioned in the text.

Lines 600 - 602: For P-wave velocity, the most valuable data available were the seismic refraction/reflection profiles (Brückl et al., 2007; Šumanovac et al., 2009) and the high-quality NAC model (Magrin and Rossi, 2020). - repetitive

The sentence has been removed.

Line 606: proved to be of high value - Why? This is not clear.

As the same sentence clearly states - for the simple reason that we had no other data in that part of the investigated area.

Lines 606 - 612: In the Fig. 5a, the northernmost profile shows the similar features as the profile interpreted by Šumanovac et al. (2009) – the crustal velocity in the SW Pannonian basin is much lower than in the Dinarides. The other profiles, located further south, also cutting across the Internal Dinarides, show that the crustal velocity in the Internal Dinarides, is generally a bit higher than in the External Dinarides with relatively quick transition to the lower seismic velocity values in the Pannonian basin (see profiles CC', DD' and EE'). - This is not a discussion. What is new, critical and relevant about the model?

This sentence has been removed.

Line 619: 6.5 km/s - should be 6.3 km/s - right?

Indeed, this was an error, and it has been corrected in the text. Thank you.

Lines 620 - 622: To test how well the newly derived 3-D model represents the true structure, we calculated the travel times for a regional earthquake recorded on representative seismic stations in the wider Dinarides area (Fig. 8) - This model validation approach should be mentioned in the introduction, the description of the modelling approach and shortly also in the abstract.

In the abstract, we have rephrased the sentence: "*The newly derived model has been compared with the simple 1D model used for routine earthquake location in Croatia, and it proved to be a significant improvement.*" to "*To validate the newly derived model, we have calculated travel times for a regional earthquake recorded on a number of seismic stations in the Dinarides area. The calculated travel times have been compared with the travel times in*

*the simple 1D model used for routine earthquake location in Croatia, and it proved to be a significant improvement.*" We haven't mentioned it in the introduction nor in the modelling approach to reduce repetition.

Lines 622 - 623: We also calculated travel times using the simple 1D model with two isotropic crust layers currently employed for routine earthquake locating in Croatia. - Reference required!

References added.

Lines 623 - 631: The 1D model's topmost layer is characterized by P-velocity of 5.8 km/s, and the deeper crustal layer has the P-wave velocity of 6.65 km/s. For the same model the uppermost mantle velocity is 8.0 km/s. We then compared the travel times from both models with the true measured travel times. We used the Pn and Pg phases of the 2020 Petrinja Mw6.4 earthquake. The location of the earthquake and the stations that recorded the wave onsets are shown in Fig. 8. For the same stations we calculated the travel times using the 1D and the new 3-D model. For travel time calculation we used the Fast Marching Method (de Kool et al., 2006) as implemented within the FMTOMO package (Rawlinson and Urvoy, 2006). - This is methodology and should be moved to a chapter "Modelling approach".

This is simply a description of the 1D model we used, and the means of calculating the travel times, so we think it should remain in this section.

Line 639: by - rephrase "between"

Thank you for your suggestion, we've made the correction in the text.

Line 640: improvement - with respect to the 1D model?

Indeed, we've rephrased the sentence to "*we can see improvement in calculated travel time accuracy when using the 3-D model (with respect to the 1-D model) for epicentral distances smaller than 50 km and over 100 km.*"

Lines 656 - 657: generally closer to the actual observed travel times for all the epicentral distances shown than those calculated using the 1D model. - This is not obvious from the figure.

The abscissa in the figure shows the difference between observed travel time and travel time calculated from 1-D (black points) and 3-D (blue crosses) models. The differences with 3-D model generally lie closer to the dashed line, representing zero, meaning they are closer to the observed travel times hence the conclusion that generally travel times for 3D model are closer to the observed ones.

Line 663: Figure 10. - Why are points of the 1d model missing at small epicentral distances?

Because for small epicentral distances there are no first arriving Pn waves refracted off the Moho interface, only reflected and they arrive later than first Pg arrival. They are possible in the 3D model, because the Moho is not horizontal as in the simple 1D model used here, but has topography.

---

## Referee Report (RR1)

**Abstract**

The abstract does not ideally present the objectives, methodology, main results and implications of the study performed. This could be improved a lot to attract more readers.

We made changes to the Abstract which hopefully now addresses the comments raised by the reviewer.

Done, but please consider comments in PDF (egusphere-2023-183-manuscript-version3.pdf).

**Introduction**

This chapter misses the clear statements of (i) which problem is at hand, (ii) why the authors are motivated to solve it (purpose of the 3D model!), (iii) which general strategy they have chosen to overcome the problem. Since the envisaged purpose of the model remains unclear in the current version of this chapter, it is impossible to assess and judge the quality of the modelling approach chosen.

In the Introduction our aim was to briefly reflect on the most important (seismic) crustal structure investigations in the Dinarides done so far. From this we draw the conclusion that there were numerous investigations of the various parts of the Dinarides but none has so far combined all these various results to create a full 3D seismic model. This would reflect on the points (i) and (ii). For the point (iii) we feel that it was already pointed out in the frame that we aim to collect the available results on the structure of the crust in the Dinarides to create a new 3D regional crustal model. Nevertheless, we reworked and rewrote the Introduction section to better emphasise all the points raised by the reviewer with special care taken to address number (ii).

Done, but please consider comments in PDF (egusphere-2023-183-manuscript-version3.pdf).

**Tectonic and geological setting**

This chapter should be shortened, i.e. reduced to information that is relevant for the understanding of the modelling approach and later discussion of results. More detailed information about lithological characteristics of the model units (Neogene, Carbonate Complex, crystalline crust) would be required to assess the reliability of the rock physical

properties of the final model.

Our view is that this section, in its current form, is necessary as it gives an overview of the tectonic-geodynamic processes that led to the complex crustal structure we see today. The aim is to introduce the reader to some of the peculiarities of the structural composition of the Dinarides and surrounding regions, such as thinning of the crust toward the Pannonian basin or thick carbonate cover and existence of deeper alluvial sedimentary cover in some areas. In our opinion this section is not too long and gives relevant background information that links the technical part of the manuscript with some of the decisions made in the model construction.

Lithological characterization in such a complex area is too broad (even significantly shortened) to fit into this article and needed detailed comparison of our results with that information is heavily out of scope of this investigation and would not in that form contribute much to the overall goal of assembling a regional seismic model. Nevertheless, in the references provided in this section the lithological properties are widely discussed and interested readers can consult those publications if needed. Furthermore, for the published results of geophysical investigations we are using, the reliability of those results has been mostly tested and discussed.

I understand the reasoning of the authors and thus can well accept their answer. The whole chapter reads fine now, though some small issues should be improved (see commented PDF).

**Data / Model construction**

The chapters "Data" and "Model construction" are not well structured (e.g., "Data" already includes information on the model construction). Furthermore, the original chapters simply fail in putting someone in the position to correctly reproduce the 3D model when using the same data - which is a primary requirement for a model to be reliable and acceptable. Therefore, for the sake of more clarity and less repetition in the text, I suggest to merge the two chapters into one called "Modelling approach" which would have the following structure: First, describe and justify the intended differentiation of the model into four discrete model units (Neogene, Carbonate Complex, crystalline crust and mantle). Then,

introduce the technicalities of the model building process (e.g., the interpolation method). Finally, present a sub-chapter for each model unit to refer to (i) the used input data sets, (ii) their processing (including equations, uncertainty assessment method) and (iii) all decisions taken by the authors including explanations (e.g., how to compensate for observational gaps; why and how to recalculate sediment thicknesses, etc.). For the later model evaluation process (Discussion), each model unit should already here come with a map showing the locations / distributions of all data that have been integrated into the modelling. Furthermore, this chapter should include all methodological aspects of the model validation process through travel time predictions (in a final sub-chapter). In general, the authors should avoid to phrase interpretations in this results chapter. In the submitted version of the manuscript, it remains unclear whether uncertainty has been assessed (calculated using equations – which?) or merely taken over from the original data sources.

Thank you for this excellent suggestion. We have now merged the two chapters as suggested and rearranged paragraphs to make them more clear. This new-merged chapter begins with the description of the data used, then we introduce kriging method, mention the need to estimate variances for handling of overlapping data, add some more explanation on the total error estimate of the model (as combination of input data error and error from interpolation itself), and then describe in detail how the results were acquired from the available data. The last paragraph of the new (merged) chapter describes how we smoothed the final model.

The authors have improved the methodological chapter significantly and it is very clear now how the model was constructed. Some minor issues are marked in the PDF.

**The main characteristics of the model**

It is not clear how useful it is to present maps showing S-wave velocity and density. In my opinion, they could be removed since they are the result of simple (empirical) P-wave velocity conversions. The resulting maps do not contribute anything to the main conclusion of this study.

Density was also calculated-interpolated, only the input data set was much smaller. We included the S-wave velocity for completeness (since we had P-wave velocity and density). This was done to facilitate easier usage of the S-velocity as a starting model for some future research (ambient noise or surface wave tomography, earthquake shaking estimation, etc.). If the reviewer feels that the S-wave model still needs to be removed, we can move S-wave figures to supplement.

They way the model parameters have been estimated is much clearer in this new version of the manuscript. I would leave the decision to the editor where to present the S-wave velocity model (main text or supplementary). As the description remains very short, it could stay in the main text as well.

**Discussion**

As an important point of the critical discussion of model limitations, the authors should comment on the fact that they have jointly interpolated P-wave velocities from active seismic profiles and those derived from gravity constrained density values. Is this reliable? There seems to be a systematic jump in velocity when crossing the boundary between the different domains. Does this mean that such abrupt changes in the interpolated maps do not reflect real differences in rock physical properties but inconsistencies due to different methods?

As the above mentioned data were the only available measurements in some areas there wasn't much choice to begin with. How reliable some of the transitions are is hard to estimate but we strongly believe that a good part of "jumps" between different domains reflects true physical properties. The models acquired from gravity data which were used in our work were taken from work of Šumanovac (2010) where the author calibrated gravity modelling based on active seismic profile parallel to one of the gravity profiles. We have added the following paragraph to the discussion chapter:

"It seems that the velocity in the Internal Dinarides, where we only had inverted gravimetric profile data available, is slightly higher than in the rest of the model. At this point, we cannot discern if it is an actual feature, or some artefact due to lower quality data. The fact is that this is a different tectonic unit, so it is not impossible that it has

different features. If we have omitted these data from interpolation, we would have even

worse results, because in that case the values would be purely extrapolated. The approach we chose gave at least some constraint to the velocity values in this part of the

model."

The authors have been more precise and more critical in describing the results in the revised manuscript.

A new, general point on this chapter: the travel time calculations have been performed to test the presented 3D model. The tests appear in the "Discussion" chapter since the authors obviously want to discuss the reliability of the model in this way. However, since they present these tests with a description of the data / techniques used and results obtained, the "Discussion" chapter loses much of its "discussing" character. This is definitely not a standard way of structuring a scientific paper, but I would leave it to the editor at this very late stage of the review process, to decide whether the authors should modify this.

**Technical corrections**

General: check and synchronize the order of figures in the paper and in the text

This was resolved.

Done.

Line 33: refer already here to the map of Figure 1 to introduce the location of the study

area

This was resolved after the comment at line 144 has been resolved.

Done.

Figure 1: add the exact position of all active seismic profiles as this is the most important

type of data for the study

The positions of the seismic profiles shown are their exact positions. We have slightly

rephrased the Figure 1 caption to point out that these are actual positions: "Blue lines

mark the positions of Alp01 and Alp02 profiles (Brückl et al., 2007) – only the full line parts

are used in the study; the red line is the position the Alp07 profile (Šumanovac et al.,

2009); black lines are the positions of gravimetric profiles GP-1 to GP-6 (Šumanovac,

2010). (…)"

Done.

Please consider also the detailed comments and suggestions in the uploaded PDF.

Line 1: rephrase to "reference"

Thank you. Rephrased.

Line 10: Abstract - Not informative, main message missing. Define "referent" seismic crustal model

Referent is used in the sense that it is the first such seismic model for the Dinarides, constructed using all the currently available results and that it will be freely available. Hence our aim is to make it a "to go" model for all future usage and that it will be refined (by us and the community) as more data becomes available.

Line 13: overlain - check language

This sentence has been rephrased to: "The good example of this are the Dinarides where thick carbonates cover older crystalline basement units and remnants of subducted oceanic crust."

Line 16: analysis - of what?

This has been rephrased to "any seismic or geological analysis"

Line 17: less - than what

Rephrased "less data coverage" to "lack of information on crustal structure"

Line 18: complete - Can a model be complete?

Rephrased to "comprehensive"

Done.

Line 21: seismic velocities (P- and S-) - How acquired? As tomography or profiles?

Neither. Seismic velocities in the model are acquired by kriging interpolation. We have added a sentence before this one: "We have used kriging interpolation to obtain the model parameters."

Done.

Line 30: all the forthcoming studies - not clear? Examples?

The sentence has been rephrased as follows: We hope that the newly assembled model will be useful for the forthcoming studies (e.g. as a starting model for seismic tomography, underlying model for earthquake simulations) which require knowledge of the crustal structure.

Done.

Line 32: wider Dinarides region - refer to Fig. for location

The wider Dinarides region here is a pretty loose term describing Dinarides proper and parts of adjacent areas such a SW Pannonian basin, Eastern Adriatic Sea, Southern Alps. After moving a paragraph from Line 144 here, it is more clear what wider Dinarides region is. Reference to Figure 1 is also present in the moved paragraph.

Done.

Line 46: virtually - meaning?

That is the exact choice of words reported in the studies by Šumanovac, 2010 and Šumanovac et al., 2016 which reported one layered crust in the Pannonian basin. We agree that the term is not exact so we removed it.

Done.

Line 78 - 79: "Even though our crustal model is focused on the Dinarides, part of it also covers the SW margin of the Pannonian basin and Adriatic Sea." - refer to map

We added reference to Figure 1.

Done.

Thank you for the suggestion, it has been added.

Done.

Line 144: The main objective of this study - Move this paragraph to the "Introduction". What is meant by "referent"? Explain the purpose of the model and give examples for later usage.

The paragraph has been moved to the beginning of the "Introduction" section and reworked to fit better in the Introduction.

What is meant by "referent" has already been answered at comment on line 10.

Done.

Lines 150 - 151: Our basic approach was to create a one-layered crust with laterally and vertically variable parameters (seismic velocities and density). - Rephrase: "...approach initially was...". This raises questions since the motivation for building the model is not well explained: - why seismic velocities and density?; - why a one-layered crust?

Thank you for pointing this out, we have modified the sentence, and now it is: "We have decided to create a one-layered crust with laterally and vertically variable parameters."

We have also added additional two sentences to explain why we chose to create a onelayered crust with the specific parameters: "The reason for a choice of one-layered crust

was a fact that not all the input data had the interpretation of intra-crustal interfaces, and

those that had, did the interpretation differently. We used seismic velocities and density

as parameters, since that is the data we had available."

Done.

Lines 161 - 161: Here, velocity and density of the carbonate layer were obtained using the same velocity (and density) data we had available for the rest of the crust. - This is not clear. How can these parameters be "obtained" if there are "no available data" (as stated in the previous sentence)?

Indeed, this is not very clear. Thank you for pointing it out. We have rephrased the sentence: "The velocity and density of the carbonate complex rocks were not interpolated separately, but were interpolated using the same input velocity (and density) data we had available for the rest of the crust."

Done.

Line 176: Figure 1. - This map should present the exact location of active seismic profiles since these are the most important and reliable source of information on vp-values.

This map is already presenting the exact locations of the profiles, and the exact extent of the other data used. We have slightly rephrased the Figure 1 caption to point out that these are actual positions: "Blue lines mark the positions of Alp01 and Alp02 profiles (Brückl et al., 2007) – only the full line parts are used in the study; the red line is the position the Alp07 profile (Šumanovac et al., 2009); black lines are the positions of gravimetric profiles GP-1 to GP-6 (Šumanovac, 2010). (...)"

Done.

Table 1 header: obtained - replace by "processed"

Thank you for the suggestion. Replaced.

Done.

Line 191: three - plus the topography/bathymetry

That is correct. The sentence has been rephrased as follows: "There are four interfaces

defined in the presented model – CRC bottom, Neogene deposits bottom, Mohorovičić discontinuity (Moho), and topography/bathymetry."

Done.

Lines 195 - 198: Since there was no velocity nor density information readily available for the Carbonate layer, its parameters were not assessed separately, as was the case for the Neogene deposits layer, but were calculated the same way as in the rest of the crust. The Carbonate bottom interface was kept in the model for the sake of completeness. - Repetitive (see above). Still the reasoning is not clear.

Removed this part from the text.

Done.

Line 209: no detailed error estimates - Unlogic: further below, the authors write "That estimate nicely fits with the error estimates reported by Bruckl et al. (2007) and Šumanovac et al. (2009) for refraction and wide-angle reflection profiles used in this study."

The point here is that there were not error estimates per-grid point like in other studies, but they were given generally, for the entire profile (±2–3 km, as mentioned a few sentences above). Error estimate of 6% of Moho depth (for each point we had!) indeed corresponds to the interval of ±2–3 km reported for profiles (generally!).

Okay.

Lines 211 - 212: include the error estimates - How are these estimates "included"?

As suggested by the reviewer we merged "Data" and "Model construction" chapters and the text is rearranged in such a way that the first mention of the total error is a combination of errors from input data and errors from interpolation itself. We hope this made the statement more clear.

Done.

Line 213: Moho model - for the European continent

Thank you for your suggestion, it has been added in the text

Done.

Line 223: depth - For clarity add "base" depth.

Thank you for your suggestion, it has been added in the text

Done.

Lines 223 - 224: The most important region of the study area - unlogic, better: "Largest volumes of ... are situated..."

Thank you for your suggestion, it has been corrected in the text.

Done.

Line 224: loose - better "unconsolidated"

Thank you for your suggestion, we've made the correction in the text.

Done.

Lines 233 - 234: We did not use interpolation in this step, in order not to introduce an additional interpolation error. - Not clear. The topic of interpolation has not been raised until this point in the manuscript. It is confusing then to read here that it is NOT used.

Here we tried to point out that we used the data in the same form as was digitised in - every 5 km along the isolines, we did not interpolate it before adding it to other data. But you are right about mentioning interpolation at this point, the sentence added a bit of confusion so we removed it from the text.

Done.

Line 234: the error estimation - Again, the authors mention an error estimation the procedure of which is not introduced and explained.

This has been improved by rearranging "Data" and "Model construction" chapters. Please

refer to replies in the beginning (after the comment on the chapters) and Line 211.

Done.

Lines 235 - 236: Since there was no error estimate beforehand, we decided to use the same percentage for Neogene deposits depth estimate. - This is not logic. Modelling errors generally arise from observational gaps, which typically increase with increasing depth. Assuming that the Moho depth errors of a continental scale model are applicable to a regional Moho may still be understandable, but applying them to a much shallower interface as well is very questionable.

As we stressed numerous times throughout the manuscript the data used come from a variety of sources some of which did not provide any error estimates. In this particular case for the Neogene deposits base depth data was taken from Saftić et al. (2003) and Matenco and Radivojević (2012) and there were no error estimates provided. We obtained the Neogene deposits depths from georeferenced isolines by digitising the maps provided in both studies. In order to provide at least some indication of error estimate we used a similar approach as was reported by Grad et al. (2009) where the authors for similar dataset and similar approach of manually digitising isolines estimate 15% error. Yes, we are aware that this is not an ideal solution but we deem that at least some error estimate is better than none in the case where the authors did not provide any error estimates or any digital data but only printed isolines. In this sense we strongly believe that our approach is logical and contributes to overall result and kriging interpolation stability.

Modelling (interpolation) uncertainties from Kriging are of-course calculated and reported as can be seen from Figures 3. and 4. in the manuscript.

Done.

Lines 238 - 239: since the basement is overlain by a thick layer of Mesozoic Carbonate rock complex. - This is not a valid explanation for missing Neogene strata. Rephrase to avoid a causal relationship (avoid "since").

Thank you for pointing this out, we have rephrased the sentence: "The area of the Dinarides does not contain a Neogene soft deposits cover, at least not of significant thickness. The basement is overlain…"

Of-course there are numerous smaller intrakarst basins filled with Neogene deposits but the data about these is scant and not easy to use. Furthermore, this data can easily be added in the future versions of this model and we encourage users to do so for smaller case studies.

Done. Thanks for explaining.

Line 245: The interface with the least data at our disposal was the Carbonate complex bottom depth. - This compares data availability for different interfaces of the model. This point requires, however, a series of maps showing input data points for each modelled unit separately.

Thank you for your suggestion, such a map has been included in the supplement, and an additional sentence has been added before Fig. 1: "The exact locations of data points used are shown in Appendix B."

Done.

Line 247: Basic Geological Maps - Why are there no geological maps mentioned in Table 1?

Thank you for your suggestion, they have been added to Table 1.

Done.

Line 251 - 252: Since the Carbonate complex represent a very distinctive layer in the Dinarides, we additionally estimated its thickness. - Reasoning of the sentence questionable. The actual reason for estimating the thickness of this unit should be the availability of thickness information (next sentence). Since the authors want to integrate all available information, this step is legitimized.

Thank you for pointing this out, we have left out that sentence.

Done.

Line 255: recalculated - This is a critical point.

- Why were thicknesses recalculated? If their present-day thickness is observed, this would be unlogic for a model that should represent present-day configurations (paleothicknesses won't be useful here)

- How have these recalculations been performed? The manuscript lacks a description that would allow a reproduction of the model with the given data.

Thank you for pointing this out. The entire paragraph was rephrased to make it more clear: "Derived values of CRC thickness were further considered in respect to the deformation styles and large-scale structural relations (e.g., Balling et al., 2021). Several regional carbonate nappe systems in the External Dinarides characterized by extensive folding and thrusting could reach a combined stacking thicknesses up to 12000 m, but thicknesses are not evenly spatially distributed. Significant variability of the CRC total thickness in the Dinarides is caused by combination of (1) initial differences in thickness due to significant paleogeographic differences along the Adria Microplate passive margin, since a total thickness of the Adriatic Carbonate Platform and thick underlying and thin overlying carbonates is in the range of 4500–8000 m (Tišljar et al., 2002; Velić et al., 2002; Vlahović et al., 2005), (2) structural position of individual nappe systems in respect to the active collision front, and (3) variable strain rates and stress orientation during the Cretaceous–Paleogene Adria–Europe collision. Nappe stacking systems in the central and southern part of the External Dinarides, where CRC is the thickest (Fig. 3c), locally incorporate up to four thrust sheets composed of different segments of the entire carbonate succession."

Done.

Line 266: Fig. 3c - Refer to figures in their consecutive order! Either move this Figure up or refer to it in the main text just after Fig. 2.

This has been fixed by rearranging "Data" and "Model construction" chapters.

Done.

Line 281: It seemed the most logical course of action, assigning the density value to the middle of the layer, to represent the entire layer. - This is a point for discussion and it is written in the style of a discussion. Hence move it there.

This merely explains how we digitised data from gravimetric profiles, and we think it should stay in the section that deals with data preparation.

Okay. Accepted.

Lines 293 - 294: we assigned the maximum error estimate (to make a conservative estimate) - Estimating errors of interface depths obtained by gravity modelling is a very complicated task because of the general non-uniqueness of gravity interpretation. Such errors mainly depend on gravity field data errors and the amounts and distribution of gravity-independent data integrated. For this reason, it does not make any sense to transfer error estimates from one gravity derived model to another. I suggest not to give any error in this case.

This is not an error estimate of interface depth, but of density value. We thought it reasonable to give a maximum error estimate from the NAC model, because authors have used the same sources for their density values.

Done. Thanks for explaining.

Line 314: 3 arc minutes in our model - This is an important characteristic of the model and the information should be given outside of brackets. Rephrase the sentence: a "regridding" has been performed.

This is merely a comment on the difference between the gridding of the SRTM15+V2.0 model and the model we derived here. The grid size of our model is mentioned later in the text. The sentence has been rephrased according to reviewers suggestion.

Done.

Line 336: 13° E and 20° E, and 42° N and 47° N. - Mention which extent has been used to present the results (maps below). Maps (below) show a slightly different coordinate range - why?

The range reported is the range we consider more reliable (hence the shaded areas in the figures). The maps are shown with a somewhat wider coordinate range, which we used for interpolation, and later marked as less reliable based on estimated errors.

Okay.

Line 371: Brocher (2008) - Please provide the relevant equations.

We omitted the equations because it would be merely copying them from the original article. We have added the following to the sentence "(eqs. 1, 3, 7, and 9 in the original article)", so the interested reader is pointed to equation numbers in the referenced article. If the reviewer insists on listing actual equations, we will add them in the supplement but we will just be cluttering the already complicated procedure.

Okay.

Lines 378 - 381: After collecting and preparing the data as described in the previous sections, the next step was interpolation. Ordinary kriging was used to interpolate model interfaces and universal kriging when interpolating the layer properties (P-velocity, density) as the layer properties are distinctly linearly dependent on depth. - This repeats information given before.

This has been fixed by rearranging "Data" and "Model construction" chapters.

Done.

Line 381: smoothed Moho discontinuity - Say how to guarantee model reproducibility. Say why: what makes a smoothed Moho advantageous compared to a non-smoothed version?

This sentence has been rephrased: "After interpolation, we filtered Moho discontinuity and layer properties with a 100-km wide Gaussian filter to smooth the transitions between

different data sources. The smoothing in case of the Neogene deposits and CRC interfaces was omitted because the data used in derivation of those interfaces came from similar sources,..."

Done.

Lines 390 - 393: Neogene deposits and Carbonate bottom depths have not been smoothed, since they have been assembled from a small number of equivalent sources. Moho discontinuity, on the other hand, is assembled from a variety of different sources, and therefore was smoothed using a Gaussian filter with a 100 km width. - Avoid repetition. Information belongs to "modelling approach" chapter.

These two sentences have been removed.

Done.

Line 398: Sava and Drava depressions - Since the locations of these depressions are not shown in the maps, nor is any other information about these structures given, referring to them in the text is not helpful.

Thank you for pointing this out. We have created a less cluttered map, which now shows the locations of the said depressions.

Done.

Line 401: That was expected, since - This is in the style of a discussion which should be avoided in the "results" part of the paper. Why was that expected? If there is a geological/tectonic reasoning that confirms the mere mathematical interpolation results, this would be a contribution to the discussion of the model quality.

We agree. The form of the sentence is somewhat imprecise and has been rephrased: "In the SW Pannonian Basin, where Neogene deposits are the thickest, the underlying carbonate layer is thin, and vice versa…"

Done.

Line 408: As corroborated before - Style. Please avoid repetition.

That sentence has been removed.

Done.

Line 417: Fig. 5 - Check figure order. Make sure you refer to Fig. 4 before referring to Fig.

5. In case, rearrange figures to fit the line of arguments.

This sentence has been removed. The reason for referring to Fig. 5 in this particular place

was to point out the mentioned feature is better seen in it.

Done.

Lines 422 - 423: one can distinctly see the areas with less data coverage as areas with

higher uncertainty. - This is indeed very difficult to see! Figure 2 shows data coverage for

all interfaces and properties in one map. However, the areas of larger uncertainty

(shadowed in maps) should be different for the different units / properties. How was the

shadowed area defined, what does it actually mean? This is not clearly explained.

We tried to estimate the areas in which the model was more accurate, and to make such

an estimate, we looked at total errors for Moho and for P-wave velocity (which were the

best constrained parameters we had). This is mentioned in the new "Modelling approach"

chapter.

Done.

Line 432: measurements - Put into a map all locations at which thickness was measured.

Such a map has been added in the supplement. Please also see the reply to the comment

at line 245.

Done.

Line 434: Figure 3 - Choose the same colour palette for maps of the same physical unit.

All interface depths (in m or km) including the Moho should have the same colour palette

(here Moho depth is presented in the same colours as vp-values in Fig. 4, which should

be avoided). Uncertainty (also in m, respectively km) could be shown by a different colour palette.

This was done on purpose. Notice that sediment and carbonate bottom depths have values going from 0 to a certain value (6 km and 15 km respectively), the chosen colour palette better shows the features (0 km is shown in white, while the maximum values are shown in black). Same is true for the errors. On the other hand, that colour palette does not adequately represent the features of the Moho, and therefore we chose the same one as for the Vp.

I would leave this decision to the editor, since this actuall regards the standards of the journal.

Line 443: separate - meaning seismic independent constraints? Or what?

Indeed, that is what it means. We have rephrased the sentence: "Given that we had no independent estimate for the velocity for the CRC we cannot discern it as a separate layer just looking at velocity values."

Done.

Line 444: In the rest of the model - meaning in the sub-sediemantary crystalline crust?

No, meaning outside of the External Dinarides. Map in Fig. 1 has been refined to better show the geological features mentioned in text, and the sentence has been rephrased to be "Outside the External Dinarides, one can see…"

Done.

Lines 449 - 452: It is hard to discern if this reflects the actual structure, or if it is the consequence of the higher uncertainty in that part of the model (the velocity here was estimated from the density values from the gravimetric profiles, given that there were no other data sources available). - This is an important point for the discussion: it is highly questionable if one should merge at all seismic velocity values from active seismic experiments with those derived from gravity-constrained density values. This may produce method-related variations not representing the actual situation. Calibrations

points would be required, i.e. locations where both types of velocity values are available. The values for seismic velocity and density derived from gravity were taken from work of Šumanovac (2010) where the author calibrated the inversion of the gravity data based on the results from active seismic profile close to and subparallel to one of the gravity profiles. For details about calibration of seismic and gravity data please see that study.

Done.

Line 460: here - where? unclear!

Indeed, it is unclear. We meant Figs. 4 and 5. It has been corrected in the text.

Done.

Line 463 - 464: using the standard P over S-velocity ratio for the upper mantle. - This requires an equation and the corresponding citation.

The sentence has been rephrased to: "...using the standard P over S-velocity ratio for the upper mantle (Vp/Vs = 1.73)."

Done.

Lines 467 - 468: and in areas where data from active seismic profiles were available (Brückl et al., 2007; Šumanovac et al., 2009) - please show the exact locations of active seismic profiles in Figure 1 for the reader to assess the relationship with the presented uncertainty.

The locations of profiles shown in Fig. 1 are their actual locations. Please see also the comment for line 176.

Done.

Line 474: Figure 4 - Abrupt changes in uncertainty (e.g. between the NAC model area and the gravity-constrained area) are visible as abrupt changes in seismic velocity. This points to systematically biased velocity distributions as a result of an unreliable merging of data.

The major contribution to the total error shown in figure 4. is the error from interpolation itself, and the errors shown in figure are expected, since we knew the amount of data we got from each of the models. As far as the velocity values go, it is the best we could do. If we did not include gravity data, we would have an area without any input data, and it would result in much larger errors in that part of the model, and even worse situation with velocity values. We are well aware that gravity data is much inferior compared to the other data sources we used, but simply had no other data to use.

Done. And thanks for the discussion of this point.

Lines 474 - 477: Velocity model depth slices and corresponding uncertainties: (a) model at a depth of 5 km, (b) uncertainty at a depth of 5 km, (c) model at a depth of 10 km, (d) uncertainty at a depth of 10 km, (e) model at a depth of 20 km, (f) uncertainty at a depth of 20 km, (g) model at a depth of 30 km, and (h) uncertainty at a depth of 30 km. - Could be shortened by less repetition. For help, just check other papers with figure panels.

Redone. Thank you for suggesting this.

"Velocity model depth slices and corresponding uncertainties for 5 km, 10 km, 20 km, and 30 km depth are shown in panels (a)(b), (c)(d), (e)(f) and (g)(h). In the panel (c) the positions of the profiles shown in Fig. 5 are marked. The areas of lower resolution are shaded, and the grey colour scale corresponds to the mantle velocity (see text for details)."

Done.

Lines 477 - 478: The areas of lower resolution are shaded - How are they defined?

This was mentioned in the new "Modelling approach" chapter. Please see the reply to the comment for "Modelling approach" section and Line 422.

Done.

Line 480: Fig. 5 - Why not showing profiles that correspond in location to the gravity profiles used as input data? What is the reasoning behind the choice of the six profiles?

We tried to pick profiles which cover the Dinarides perpendicular to their axis, because that is where the most interesting features can be seen. Moreover, these profiles are actually close to the locations of the gravity profiles and can be easily compared. Also, we chose one profile along the Dinarides' axis, to show how the features change along strike.

Done.

Line 480: b - c

In one of the working versions of text, this was indeed panel b, we haven't noticed that we didn't make the change in all places it was mentioned. The error has been corrected in the text.

Done.

Line 490: almost perfectly - rephrase to "in large parts of the profile"

Thank you for your suggestion, we've made the correction in the text.

Done.

Line 490: value - rephrase to "isoline"

Thank you for your suggestion, we've made the correction in the text.

Done.

Lines 515 - 516: This can be linked with the remnants of the subducted lithosphere and the ongoing underthrusting or lithospheric delamination (see e.g., discussion in Stipčević et al. 2020) - This is interpretation to be removed from this "results" chapter!

This sentence has been removed, because it carries the same information as the one in comment for Line 536.

Done.

Line 529: Figure 5 - The colour for lowest velocity of the crust is actually black due to the

tightness of isolines. Therefore, these areas cannot be differentiated from the black regions of highest velocity in the mantle. The figure is not unambiguous and requires respective modifications.

Thank you for your suggestion, a modified figure has been added in the text.

Done.

Thank you for your suggestion, we've made the correction in the text.

Done.

The error has been corrected in the text.

Done.

Line 529: AA' - "A" and "A'" should be shown at the ends of the profiles in the figure!

Thank you for your suggestion, a modified figure has been added in the text.

Done.

Thank you for your suggestion, we've made the correction in the text.

Done.

Lines 536 - 544: These positive anomalies in the work of Belinić et al. (2020) have been interpreted as a signal of the subducting Adria Microplate. Our model is mere interpolation of what was already known, but perhaps what we see here is part of the Adria crust being dragged along the uppermost part of the mantle being subducted below the Dinarides. As can be seen in the Fig. 5a, which is crossing into the SW Pannonian basin, the crustal

velocity in that part is much lower than in the Dinarides, a feature also observed by Šumanovac et al. (2009). Perhaps the lower velocity is a feature of the Pannonian crust, whereas the relatively higher crustal velocity is a feature of the Adria crust. To make a definitive conclusion, more investigation should be performed. - Move to Discussion chapter!

Thank you for your suggestion, this whole paragraph has been moved to the "Discussion" chapter.

Done.

Line 559: The areas of lower resolution are shaded, and the gray color scale corresponds to the mantle density (see text for details) - It is hardly possible to discriminate between shaded and gray areas in the map. Suggestion: replace shading through hatched overprint.

Thank you for the suggestion, we have improved this figure with different shading. We have also updated the figures representing the other parameters.

Done.

Line 569: S-wave velocity at four depths. - As it is a mere linear conversion from vp, does it provide any new and relevant information?

We included the S-wave velocity for completeness (since we had P-wave velocity and density). This was done to facilitate easier usage of the S-velocity as a starting model for some future research (ambient noise or surface wave tomography, earthquake shaking estimation, etc.).

Accepted.

Lines 580 - 581: the need for the more complete seismic model of the Dinarides - From what does this need arise? Who would need this model and for which purpose(s) could it be used?

The knowledge of crustal structure is needed in any seismological study that deals with

waves propagation (tomography, earthquake location, source inversion, etc.). One could easily state that having a good starting (referent) model is of utmost importance in any seismological study. Moreover, the existence of a comprehensive model readily available in digital form greatly simplifies the preparation of input data for any such study.

Done.

Line 585: enough - What makes it be "enough"?

It never is "enough". It has been rephrased to "most".

Done.

Lines 585 - 586: It is mainly based on Stipčević et al. (2020) study and our model reflect all the features noted therein. - This is repetitive and trivial.

The sentence has been removed.

Done.

Lines 587 - 591: As shown in Fig. 5f the Moho deepens going from the NW to the SE along the main axis of the Dinarides. Also, from the profiles in Figs. 5a–e the change in Moho depth going from the Adriatic Sea towards the Dinarides mountain chain is much more gradual than on the other side, going from the Dinarides towards the SW Pannonian basin, where this change is much more abrupt, almost step-like. - This is a repetition of results, while here the reliability and relevance of these outcomes should be discussed.

The sentences have been replaced with: "It confirms what we already know about Moho in the Dinarides, but now it is presented as a comprehensive, ready-to-use model."

Done.

Line 591: The same feature has been observed by Šumanovac et al. (2009) and Šumanovac (2010). - So, does the model provide any new results at all? What would these new results be?

The model has been assembled from existing results; it cannot provide "new" results, but

now it is presented as a comprehensive, ready-to-use model. The sentence has been removed.

Done.

Lines 592 - 593: In the case of Neogene deposits thickness the data used came from two sources (Saftić et al., 2003 and Matenco and Radivojević, 2012) - repetitive

The sentence has been removed.

Done.

Lines 595 - 596: this parameter was adequately presented in our model. - What is the evidence for this statement?

We agree that this statement was a bit subjective. Therefore we rephrased it to the following:

"For the Neogene deposit thickness we used manually digitized maps, therefore having less precise data, but which were originally created from a high number of active seismic profiles and thus strengthening our confidence that this parameter was adequately presented in our model."

Done.

Lines 596 - 597: Sava and Drava depressions - Again, without showing these structures, readers that are not familiar with the region will not understand this.

Map in Fig. 1 has been refined to better show the geological features mentioned in the text.

Done.

Lines 600 - 602: For P-wave velocity, the most valuable data available were the seismic refraction/reflection profiles (Brückl et al., 2007; Šumanovac et al., 2009) and the highquality NAC model (Magrin and Rossi, 2020). - repetitive

The sentence has been removed.

Done.

Line 606: proved to be of high value - Why? This is not clear.

As the same sentence clearly states - for the simple reason that we had no other data in that part of the investigated area.

Done.

Lines 606 - 612: In the Fig. 5a, the northernmost profile shows the similar features as the profile interpreted by Šumanovac et al. (2009) – the crustal velocity in the SW Pannonian basin is much lower than in the Dinarides. The other profiles, located further south, also cutting across the Internal Dinarides, show that the crustal velocity in the Internal Dinarides, is generally a bit higher than in the External Dinarides with relatively quick transition to the lower seismic velocity values in the Pannonian basin (see profiles CC', DD' and EE'). - This is not a discussion. What is new, critical and relevant about the model?

This sentence has been removed.

Done.

Line 619: 6.5 km/s - should be 6.3 km/s - right?

Indeed, this was an error, and it has been corrected in the text. Thank you.

Done.

Lines 620 - 622: To test how well the newly derived 3-D model represents the true structure, we calculated the travel times for a regional earthquake recorded on representative seismic stations in the wider Dinarides area (Fig. 8) - This model validation approach should be mentioned in the introduction, the description of the modelling approach and shortly also in the abstract.

In the abstract, we have rephrased the sentence: "The newly derived model has been compared with the simple 1D model used for routine earthquake location in Croatia, and

it proved to be a significant improvement." to "To validate the newly derived model, we

have calculated travel times for a regional earthquake recorded on a number of seismic

stations in the Dinarides area. The calculated travel times have been compared with the

travel times in the simple 1D model used for routine earthquake location in Croatia, and

it proved to be a significant improvement." We haven't mentioned it in the introduction nor

in the modelling approach to reduce repetition.

I appreciate that this important step is mentioned in the abstract now. Instead of describing the technique in the discussion chapter, however, I would move it to the "modelling approach" chapter (see also below).

Lines 622 - 623: We also calculated travel times using the simple 1D model with two isotropic crust layers currently employed for routine earthquake locating in Croatia. - Reference required!

References added.

Done.

Lines 623 - 631: The 1D model's topmost layer is characterized by P-velocity of 5.8 km/s, and the deeper crustal layer has the P-wave velocity of 6.65 km/s. For the same model the uppermost mantle velocity is 8.0 km/s. We then compared the travel times from both models with the true measured travel times. We used the Pn and Pg phases of the 2020 Petrinja Mw6.4 earthquake. The location of the earthquake and the stations that recorded the wave onsets are shown in Fig. 8. For the same stations we calculated the travel times using the 1D and the new 3-D model. For travel time calculation we used the Fast Marching Method (de Kool et al., 2006) as implemented within the FMTOMO package (Rawlinson and Urvoy, 2006). - This is methodology and should be moved to a chapter "Modelling approach".

This is simply a description of the 1D model we used, and the means of calculating the

travel times, so we think it should remain in this section.

I would still argue for moving this paragraph to the "modelling approach" chapter (see also the comment in the PDF), but I would leave the final decision to the editor.

Line 639: by - rephrase "between"

Thank you for your suggestion, we've made the correction in the text.

Done.

Line 640: improvement - with respect to the 1D model?

Indeed, we've rephrased the sentence to "we can see improvement in calculated travel time accuracy when using the 3-D model (with respect to the 1-D model) for epicentral distances smaller than 50 km and over 100 km."

Done.

Lines 656 - 657: generally closer to the actual observed travel times for all the epicentral distances shown than those calculated using the 1D model. - This is not obvious from the figure.

The abscissa in the figure shows the difference between observed travel time and travel time calculated from 1-D (black points) and 3-D (blue crosses) models. The differences with 3-D model generally lie closer to the dashed line, representing zero, meaning they are closer to the observed travel times hence the conclusion that generally travel times for 3D model are closer to the observed ones.

Can you still comment on why there is a trend of 3D model points to show negative time difference, while 1D model points mostly show positive values (for both Pg and Pn phases)?

Line 663: Figure 10. - Why are points of the 1d model missing at small epicentral distances?

Because for small epicentral distances there are no first arriving Pn waves refracted off the Moho interface, only reflected and they arrive later than first Pg arrival. They are possible in the 3D model, because the Moho is not horizontal as in the simple 1D model used here, but has topography.

Thanks for the explanations!

[revised manuscript text omitted]

---

## Author Response (AR2)

Dear Editor,

We accepted all the suggestions and made the changes accordingly. Below please find our answers and annotated manuscript detailing larger changes. The comment-replies are structured as specified below.

We hopefully corrected most of the language inconsistencies and grammar mistakes. Furthermore, we removed part of the text that were redundant. All the changes done are document in the marked manuscript.

Kind regards,
Authors

In red - reviewers initial comment
In black - authors reply
In green - reviewers final reply
in blue – Editors final comment

**Comment #1**

It is not clear how useful it is to present maps showing S-wave velocity and density. In my opinion, they could be removed since they are the result of simple (empirical) P-wave velocity conversions. The resulting maps do not contribute anything to the main conclusion of this study.

Density was also calculated-interpolated, only the input data set was much smaller. We included the S-wave velocity for completeness (since we had P-wave velocity and density). This was done to facilitate easier usage of the S-velocity as a starting model for some future research (ambient noise or surface wave tomography, earthquake shaking estimation, etc.). If the reviewer feels that the S-wave model still needs to be removed, we can move S-wave figures to supplement.

The way the model parameters have been estimated is much clearer in this new version of the manuscript. I would leave the decision to the editor where to present the S-wave velocity model (main text or supplementary). As the description remains very short, it could stay in the main text as well.

We agree with the reviewer and would leave the S-model in the main text if the Editor agrees.

C.B.: I agree with leaving the Vs model as it may be useful for readers and for future use.

Agree. Left the Vs model in the manuscript.

**Comment #2**

As an important point of the critical discussion of model limitations, the authors should comment on the fact that they have jointly interpolated P-wave velocities from active seismic profiles and those derived from gravity constrained density values. Is this reliable? There seems to be a systematic jump in velocity when crossing the boundary between the different domains. Does this mean that such abrupt changes in the interpolated maps do not reflect real differences in rock physical properties but inconsistencies due to different methods?

As the above mentioned data were the only available measurements in some areas there wasn't much choice to begin with. How reliable some of the transitions are is hard to estimate but we strongly believe that a good part of "jumps" between different domains reflects true physical properties. The models acquired from gravity data which were used in our work were taken from work of Šumanovac (2010) where the author calibrated gravity modelling based on active seismic profile parallel to one of the gravity profiles. We have added the following paragraph to the discussion chapter:
"It seems that the velocity in the Internal Dinarides, where we only had inverted gravimetric profile data available, is slightly higher than in the rest of the model. At this point, we cannot discern if it is an actual feature, or some artefact due to lower quality data. The fact is that this is a different tectonic unit, so it is not impossible that it has different features. If we have omitted these data from interpolation, we would have even worse results, because in that case the values would be purely extrapolated. The approach we chose gave at least some constraint to the velocity values in this part of the model."

The authors have been more precise and more critical in describing the results in the revised manuscript. A new, general point on this chapter: the travel time calculations have been performed to test the presented 3D model. The tests appear in the "Discussion" chapter since the authors obviously want to discuss the reliability of the model in this way. However, since they present these tests with a description of the data / techniques used and results obtained, the "Discussion" chapter loses much of its "discussing" character. This is definitely not a standard way of structuring a scientific paper, but I would leave it to the editor at this very late stage of the review process, to decide whether the authors should modify this.

We agree with the reviewer that it's not a standard way of structuring but feel that leaving it within the discussion section greatly improves the readability of the manuscript and improves the overall discussion by including model testing within discussion on model characteristics.

C.B.: I agree with the reviewer and made a suggestion in the annotated PDF of the manuscript.

Done. We transferred the part of model testing to a new chapter named "Model testing" and placed it between "Results" and "Discussion" sections.

**Comment #3**

Line 434: Figure 3 - Choose the same colour palette for maps of the same physical unit. All interface depths (in m or km) including the Moho should have the same colour palette (here Moho depth is presented in the same colours as vp-values in Fig. 4, which should be

avoided). Uncertainty (also in m, respectively km) could be shown by a different colour palette.

This was done on purpose. Notice that sediment and carbonate bottom depths have values going from 0 to a certain value (6 km and 15 km respectively), the chosen colour palette better shows the features (0 km is shown in white, while the maximum values are shown in black). Same is true for the errors. On the other hand, that colour palette does not adequately represent the features of the Moho, and therefore we chose the same one as for the Vp.

I would leave this decision to the editor, since this actuall regards the standards of the journal.

We stand by our first answer as this colour scheme greatly enhances visual representation of different values within the model.

C.B.: Here again I agree with the reviewer. Please, replot those figures in a more consistent manner. It will be easier for the readers if you employ a more "traditional" way of plotting your results.

Agree. Changed as requested by the Reviewer.

**Comment #4**

Lines 620 - 622: To test how well the newly derived 3-D model represents the true structure, we calculated the travel times for a regional earthquake recorded on representative seismic stations in the wider Dinarides area (Fig. 8) - This model validation approach should be mentioned in the introduction, the description of the modelling approach and shortly also in the abstract.

In the abstract, we have rephrased the sentence: "The newly derived model has been compared with the simple 1D model used for routine earthquake location in Croatia, and it proved to be a significant improvement." to "To validate the newly derived model, we have calculated travel times for a regional earthquake recorded on a number of seismic stations in the Dinarides area. The calculated travel times have been compared with the travel times in the simple 1D model used for routine earthquake location in Croatia, and it proved to be a significant improvement." We haven't mentioned it in the introduction nor in the modelling approach to reduce repetition.

I appreciate that this important step is mentioned in the abstract now. Instead of describing the technique in the discussion chapter, however, I would move it to the "modelling approach" chapter (see also below).

On this one, we feel that it should stay in the "Discussion" section. The testing of the model does not have any connection with how the model was assembled i.e. "Modelling approach" and is much more useful when discussing the performance of the new model.

C.B.: Please, see my comment above and in the annotated PDF of the paper regarding the discussion section.

Done as requested by the Reviewer and Editor. Created new chapter named "Model testing" and moved the part of the discussion there.

**Comment #5**

Lines 623 - 631: The 1D model's topmost layer is characterized by P-velocity of 5.8 km/s, and the deeper crustal layer has the P-wave velocity of 6.65 km/s. For the same model the uppermost mantle velocity is 8.0 km/s. We then compared the travel times from both models with the true measured travel times. We used the Pn and Pg phases of the 2020 Petrinja Mw6.4 earthquake. The location of the earthquake and the stations that recorded the wave onsets are shown in Fig. 8. For the same stations we calculated the travel times using the 1D and the new 3-D model. For travel time calculation we used the Fast Marching Method (de Kool et al., 2006) as implemented within the FMTOMO package (Rawlinson and Urvoy, 2006). - This is methodology and should be moved to a chapter "Modelling approach".

This is simply a description of the 1D model we used, and the means of calculating the travel times, so we think it should remain in this section.

I would still argue for moving this paragraph to the "modelling approach" chapter (see also the comment in the PDF), but I would leave the final decision to the editor.

As before we feel that describing this 1D model in the discussion gives better overall covering when testing the model. If we described this in the "Modeling approach" it would be lost until we got to the point where we actually use this model. Also, we are not using this 1D model to create a new model but to test the performance of the new model hence it should be in the discussion.

C.B.: I agree with the reviewer.

Done. See our answer to Comments #2 and #4.